A pilot study about microplastics and mesoplastics in an Antarctic glacier:
the role of aeolian transport
Miguel González-Pleiter[1,2]†, Gissell Lacerot[3], Carlos Edo[1], Juan Pablo-Lozoya[4], Francisco
Leganés[2], Francisca Fernández-Piñas[2], Roberto Rosal[1], Franco Teixeira-de-Mello[5]†
[1]Department of Analytical Chemistry, Physical Chemistry and Chemical Engineering,
University of Alcala, Alcalá de Henares, E-28871 Madrid, Spain
[2]Departament of Biology, Faculty of Sciences, Universidad Autónoma de Madrid,
Cantoblanco, E-28049 Madrid, Spain
[3]Ecología Funcional de Sistemas Acuáticos, Centro Universitario Regional del Este
(CURE), Universidad de la República, Ruta nacional Nº9 y ruta Nº15, 27000 Rocha,
Uruguay
[4]Centro Interdisciplinario de Manejo Costero Integrado del Cono Sur (C-MCISur), Centro
Universitario Regional del Este (CURE), Universidad de la República, Tacuarembó entre
Av. Artigas y Aparicio Saravia, 20000 Maldonado, Uruguay
[5]Departamento de Ecología y Gestión Ambiental, Centro Universitario Regional del Este
(CURE), Universidad de la República, Tacuarembó entre Av. Artigas y Aparicio Saravia,
20000 Maldonado, Uruguay
†Corresponding authors:
Miguel González-Pleiter, email: mig.gonzalez@uam.es
Franco Teixeira-de-Mello, email: frantei@fcien.edu.uy

**Abstract**
Plastics have been found in several compartments in Antarctica. However, there is
currently no evidence of their presence in Antarctic glaciers. Our pilot study investigated
plastic occurrence on two ice surfaces (one area close to Uruguay lake and another one
close to Ionosferico lake) that constitute part of the ablation zone of Collins Glacier (King
George Island, Antarctica). Our results showed that expanded polystyrene (EPS) was
ubiquitous ranging from 0.17 to 0.33 items m$^{-2}$ whereas polyester was found only on the
ice surface close to Uruguay lake (0.25 items m$^{-2}$). Furthermore, we evaluated the daily
changes in the presence of plastics in these areas in the absence of rainfall to clarify the
role of the wind in their transport. We registered an atmospheric dry deposition rate
between 0.08 items m$^{-2}$ day$^{-1}$ on the ice surface close to Uruguay lake and 0.17 items m$^{-2}$
$^2$ day$^{-1}$ on the ice surface close to Ionosferico lake. Our pilot study is the first report of
plastic pollution presence in an Antarctic glacier, possibly originated from local current
and past activities, and the first to assess the effect of wind in its transport.

**Introduction**

The cryosphere is the frozen water part of the Earth system that consists of areas in which the temperatures are below 0°C for at least part of the year (NOAA, 2019). Most of the cryosphere in terms of volume of ice is in Antarctica. Despite the increasing rate of ice loss during last decades (Rignot et al., 2019), it has been estimated that the Antarctic cryosphere holds around 90% of Earth's ice mass (Dirscherl et al., 2020). Furthermore, the Antarctic cryosphere represents the majority of the world's freshwater, representing the largest freshwater ecosystem on the planet (Shepherd et al., 2018).

Plastics, especially microplastics (plastic items < 5 mm long; MPs), have been detected in several specific locations of the cryosphere including mountain glaciers (Ambrosini et al., 2019; Cabrera et al., 2020; Materić et al., 2020), snow (Bergmann et al., 2019; Österlund et al., 2019) and sea ice (Geilfus et al., 2019; Kelly et al., 2020; La Daana et al., 2020; Obbard et al., 2014; Peeken et al., 2018; von Friesen et al., 2020). The occurrence of MPs in snow ranged from 0 to $1.5 \times 10^5$ MP $L^{-1}$ of melted snow (Bergmann et al., 2019), although it should be noted that a part of this study was conducted near urban areas. Regarding sea ice, concentrations of up to $1.2 \times 10^4$ MP $L^{-1}$ have been reported, although there are large differences between studies even from the same region (Peeken et al., 2018; von Friesen et al., 2020). The use of different units in reporting MP concentrations in mountain glaciers such as the number of items per mass of ice weight (78.3 ± 30.2 MPs $kg^{-1}$ of sparse and fine supraglacial debris; Ambrosini et al., 2019) and mass of MPs per volume (0 to 23.6 ± 3.0 ng of MPs $mL^{-1}$; Materić et al., 2020), makes comparisons between studies difficult (101.2 items $L^{-1}$; Cabrera et al., 2020). Regarding the shape of the MPs found in the cryosphere, fibers seem to be dominant in mountain glaciers (65 %) and sea ice (79 %), followed by fragments (Ambrosini et al., 2019; La Daana et al., 2020). Concerning the size of MPs, it has been reported a broad size distribution in sea ice, with 67 % of MPs in the 500-5000 μm range (La Daana et al., 2020). Other studies found lower sizes, however, with significant amounts (up to 90 %) of MPs smaller than 100 μm in snow and sea ice (Ambrosini et al., 2019; Bergmann et al., 2019; Bergmann et al., 2017; Kelly et al., 2020; Peeken et al., 2018). The differences between these studies may be due to the different analytical methods used, particularly methodologies such as micro Fourier transform infrared spectroscopy (μFTIR, which can identify smaller sized MPs). In general, the presence of plastics > 5mm has not been reported in the cryosphere, probably because they occur at lower concentrations and evade detection. μFTIR revealed that polyethylene terephthalate (PET), polyamide (PA), polyester (PE), varnish (acrylates/polyurethane), several synthetic rubbers, polypropylene (PP), and polyurethane (PU) are the most common types of MPs in the cryosphere (Ambrosini et al., 2019; Bergmann et al., 2019; Bergmann et al., 2017; La Daana et al., 2020; Materić et al., 2020; Obbard et al., 2014; Peeken et al., 2018). The sources of MPs detected in the cryosphere, however, remain poorly understood. It has been suggested that they could be transported by the wind before being deposited by both wet and dry deposition in remote areas such as polar regions (Halsband and Herzke, 2019). In fact, it has been reported that air masses can transport MPs through the atmosphere over distances of

at least 100 km and that they can be released from the marine environment into the
atmosphere by sea-spray (Allen et al., 2020; Allen et al., 2019; González-Pleiter et al.,
2020a).
So far, plastics have been found in specific parts of the cryosphere (mountain glacier,
snow, and sea ice) and Antarctica (seawater, freshwater, sediments, and organisms). We
hypothesize that plastics have also reached freshwater glaciers in Antarctica and that
atmospheric dry deposition plays a crucial role in this process. To test this hypothesis,
we carried out a pilot study to investigate the presence of plastics on two ice surfaces
(one area close to Uruguay lake and another one close to Ionosferico lake) that
constitute part of the ablation zone of Collins Glacier in Maxwell Bay in King George
Island (Antarctica). Furthermore, the daily changes in the presence of plastics in these ice
surfaces was evaluated in the absence of rainfall, to clarify the role of wind in their
transport.
**Materials and Methods**
2.1 Study area
Collins Glacier is located on the northeast of Fildes Peninsula (King George Island,
Antarctica; Figure 1A) and has a total surface area of 15 km$^2$ (Simoes et al., 2015). Our
study was carried out on the ice surface of the glacier ablation areas close to two lakes
(Uruguay or Profound, and Ionosferico) in Maxwell Bay (Figure 1B). Uruguay lake (S 62°
11' 6.54'', O 58° 54' 42.23'') is located in the proximity of the Artigas Antarctic Scientific
Base and its access road (~300 m) is subjected to human transit (Figure 1B). The distance
from the shoreline to Uruguay lake is ~366 m. The lake is used for drinking and domestic
water supply. The glacier surface studied in this lake covered 1680 m$^2$. Ionosferico lake
(62° 11' 59.41'', O 58° 57' 44.17'') is located ~600 m from Artigas Base and has minimal
human activity. The distance from the shoreline to Ionosferico lake is ~694 m. The glacier
surface studied in this lake covered 537 m$^2$ (Figure 1B). It should be noted that there
were no visible footpaths through or nearby the glacier surfaces of both lakes during the
duration of our study (except our own footprints).
2.2 Experimental assessment of plastic concentration
To evaluate the concentration of plastics, twelve squares were marked on the ice
surface close to Uruguay lake (Figure 1C) and six squares on the ice surface close to
Ionosferico lake (Figure 1D), which constitute part of the ablation zone of Collins Glacier,
on 18/2/2020. The first square of 1m$^2$ on the ice surface close to each lake was randomly
marked. After that, the rest of the squares of 1m$^2$ were distributed every ten meters
covering the entire ice surface in each lake (Figure 1E). All items visually resembling
plastic (suspected plastic) inside the squares were registered (Figure 1F). It should be
noted that our sampling strategy excluded the plastics non-detectable by the naked eye
(i.e. small plastics such as fibers). Thus, we probably underestimated the concentration
of small plastics on the ice surface.
2.3 Experimental assessment of atmospheric dry deposition of plastics
After the initial sampling, we selected six squares on the ice close to each lake for
subsequent daily monitoring. Additional sampling was performed every twelve hours for
two days (18/02/2020 and 20/02/2020) after the initial sampling. No rainfall occurred
during the duration of the experiment.
2.4 Characterization and identification of plastics
Every item visually resembling plastic detected in the squares was collected with
stainless-steel tweezers, placed into glass bottles, and stored at 4 °C until analysis. All
collected items were photographed, measured and their composition was identified by
ATR-FTIR using an Agilent Cary 630 FTIR spectrometer or by µFTIR on a Perkin-Elmer
Spotlight 200 Spectrum Two apparatus equipped with a MCT detector (depending on
the size of the item). The spectra were taken using the following parameters in micro-
transmission mode: spot 50 µm, 32 scans, and spectral range 550-4000 cm$^{-1}$ with 8 cm$^{-1}$
$^{1}$ resolution. The spectra were processed using Omnic software (Thermo Fisher). Items
with matching values > 60% were considered plastic materials. The results of
concentration and atmospheric dry deposition of plastics reported in this study include
only items positively identified as plastics according to the FTIR analysis and were
expressed as number of items per surface unit and items per surface unit and day
respectively.
2.5 Prevention of procedural contamination
To avoid sample contamination, all materials used were previously cleaned with MilliQ
water, wrapped in aluminum foil, and heated to 300 °C for 4 h to remove organic matter.
The use of any plastic material during sampling was avoided. Furthermore, possible
contamination from our clothes was controlled throughout the sampling, by checking
fibers and fragments extracted from the clothes against the MPs and MePs found in the
samples, and by positioning us against the wind during sampling. Given their size,
plastics found in this study were detected by the naked eye and their traceability could
be easily maintained during quantification and identification of the samples.
**Results and discussion**
3.1 Characterization and identification of the plastics
In total, 45 items preliminarily identified as plastics were collected, of which 29 items
were confirmed as plastic by FTIR or µFTIR analyses (matching > 60%).  The size of
plastics ranged from 2292 to 12628 µm length and from 501 to 11334 µm width (Figure
2A). According to their size, 13 mesoplastic items (plastic items between 5-25 mm long;
MeP) and 3 MP items were found on the ice close to Uruguay lake, and 12 MeP items
and 1 MP item on the ice close to Ionosferico lake (Figure 2B). Meso and MPs
(hereinafter referred to as plastics) of expanded polystyrene (EPS) were found on the
ice close to both lakes: 8 plastic items on the ice close to Uruguay lake and 13 plastic
items on the ice close to Ionosferico lake (Figure 2B, C, and D). Polyester (n = 7 items;
Figure 2B, E, and F) and polyurethane (n = 1 item; Figure 2B, G and H) items were present
only on the ice close to Uruguay lake. It should be noted that spectra of the polyester
(Figure 2F) showed a high similarity with alkyd resin, a thermoplastic polyester widely
used in synthetic paints.
3.2 Plastic concentration
EPS items were ubiquitous on the ice with concentrations ranging from 0.17 items m$^{-2}$
on the ice close to Uruguay lake to 0.33 items m$^{-2}$ on the ice close to Ionosferico lake
(Table S1). The concentration of polyester, which was found only on the ice close to
Uruguay lake, was 0.25 items m$^{-2}$ (Table S1). Polyurethane items were not observed in
Ionosferico lake (Table S1).
3.3 Atmospheric dry deposition of plastics
The dry deposition rate of EPS was 0.08 EPS items m$^{-2}$ day$^{-1}$ and 0.17 EPS items m$^{-2}$ day$^{-}$
$^{1}$ on the ice close to Uruguay and Ionosferico lakes, respectively (Table S2 and Figure 3).
Polyester was only deposited on the ice close to Uruguay lake at a rate of 0.08 items m$^{-}$
$^{2}$ day$^{-1}$. Polyurethane items were not observed in Ionosferico lake during the duration
of the experiment (Table S2). The plastics deposited on the ice of Ionosferico lake during
the experiment were exclusively EPS (Table S2 and Figure 3).
**Discussion**
The presence of plastics has been documented in different places in Antarctica: marine
surface waters (Cincinelli et al., 2017; Isobe et al., 2017; Jones-Williams et al., 2020;
Lacerda et al., 2019; Suaria et al., 2020), marine sediments (Cunningham et al., 2020;
Munari et al., 2017; Reed et al., 2018), zooplankton samples from ocean water (Absher
et al., 2019), marine benthic invertebrates (Sfriso et al., 2020), Antarctic Collembola
(Bergami et al., 2020b), penguins (Bessa et al., 2019), seabirds (Ibañez et al., 2020) and
freshwater (González-Pleiter et al., 2020b). However, there was only one study showing
the occurrence of plastics in the Antarctic cryosphere, which was carried out on sea ice
(Kelly et al., 2020). Thus, this is the first report on the presence of MPs and MePs in
Antarctic freshwater glaciers. Furthermore, our findings provide an insight into the role
of wind in the transport of this material.
In this sense, winds (especially high-speed ones) appear to be a key element in the
transport of plastics to Antarctic glaciers. The prevailing winds in the study area (Figure
1B) blow predominantly from the west (Figure 4A). However, strong winds (Figure 4B),
wind gusts (Figure 4C), and strong wind gusts (Figure 4D) blow mainly from the east and
southeast directions, and could be responsible for the spreading of plastics from the
different origins to the surface of the glacier ablation areas. These strong winds would
explain the presence of MePs despite their size (Figure 2A). In fact, the low density of
the MePs found (mainly EPS; Figure 2B) would have allowed their easy dispersion by
wind.
Our results on the dry deposition of plastics support the hypothesis that the role of the
wind is essential for the transport of MPs and MePs in (and among) different areas of

Antarctica. The dry deposition of plastics (Table S2) was closely related to the wind regimes during the study period (Figure S1). Based on information available on the meteorological conditions during the study dates (18/02/2020 - 20/02/2020) in Villa Las Estrellas (Figure S1A), which is located near the Artigas Beach (Figure S1B), the wind blew from the northeast veering to the south with a speed between 10 and 30 km/h (Figure S1A). These wind conditions suggest a possible link with marine environment, which can act as a source of plastics (Allen et al., 2020), and potentially explain the presence of plastics on the glacier ablation areas. However, considering the low intensity of the winds recorded during those days (Figure S1A) and the presence of MePs, it is also possible that the predominant high-speed winds transported MePs from other adjacent areas of the Fildes Peninsula to the vicinity of the lakes, in the days prior to our study (Figure 4B, C, and D)  and then, the milder winds registered during the sampling days (Figure S1A)  deposited these MePs on the ice.

The chemical composition of the plastics found (Figure 2D, F, and H) supports the fact that the source of the plastics could be of marine and/or land-based origin. The types of plastics found (Figure 2B) are related to human activities in the Fildes Peninsula that could generate plastic debris such as tourism, leaks in waste management at scientific bases or the presence of abandoned infrastructures. Considering the location of Collins Glacier and the main human activities on the Fildes Peninsula (e.g. airfield, scientific bases), the prevailing winds from the west could have transported small and lightweight plastics to the study area. In fact, EPS is widely used in packaging and as insulation material in old buildings in this area and polyester is also a component of old buildings paints. In the same way, some of these plastics could be released from the marine environment to Artigas beach area and, then, be transported by the wind to the glaciers. In this sense, polyurethane MePs (which are similar to those found in this work) have already been reported in sea surface waters in the Antarctic (Jones-Williams et al., 2020) and EPS MePs have been found on Artigas beach (Laganà et al., 2019). These findings highlight a potential threat to the fragile Antarctic ecosystem, since the presence of these plastics (e.g. polystyrene particles) has been shown to affect Antarctic biota (Bergami et al., 2019; Bergami et al., 2020a).

The role of the atmospheric dry deposition on the presence of plastics on glaciers is supported by recent studies suggesting that MPs can be transported, up to hundreds of kilometres, through the atmosphere before being deposited (González-Pleiter et al., 2020a). Our results showed that the atmospheric deposition of plastics on glaciers is still low, with figures between two and four orders of magnitude lower than values reported in populated areas (Brahney et al., 2020; Cai et al., 2017; Dris et al., 2016; Klein and Fischer, 2019; Roblin et al., 2020; Wright et al., 2020). Our results also show that plastic pollution, even if only in small quantities, reaches remote areas with few human settlements. The occurrence of plastic pollution in Antarctica represents the spreading of anthropogenic pollutants in the last pristine environment on the Earth. Further research is needed then to elucidate the occurrence, sources, fate, and impact of plastics in such remote places.


Taken together, our research indicates that human activities in sensitive remote areas
such as Antarctica leave a footprint that includes plastic pollution. Since the early reports
of litter pollution on the seafloor (Dayton and Robilliard, 1971) and ,subsequently, on
beaches and seabirds of Antarctica (Convey et al., 2002; Creet et al., 1994; Fijn et al.,
2012; Lenihan et al., 1990; Sander et al., 2009) the handling of waste has been improved
by the implementation of the Antarctic Treaty System, Annex III 'Waste Disposal and
Waste Management'. The Treaty forces to remove all plastic from Antarctica, with the
only exception of plastics that can be incinerated without producing harmful emissions
(Antarctic Treaty Secretariat, 1998). However, once plastics are broken down into
smaller fractions and dispersed throughout the continent and nearby waters,
management measures become very difficult to address, as evidenced by our data.
Thus, a more rigorous management of plastics is essential for preserving a clean
environment within the Treaty Area (Zhang et al., 2020).


**Conclusion**
This is the first report of the presence of both MePs and MPs in an Antarctic glacier,
which were probably transported by wind from local sources such as beach areas. In
total, three types of plastics (EPS, PU and polyester) were found on two glacier surfaces
that constitute part of the ablation zone of Collins Glacier (King George Island,
Antarctica). EPS was ubiquitous in the two glacier surfaces studied. Our study showed
that the management of plastic contamination in Antarctica should be improved,
focusing on the waste generated by current and past anthropogenic activities that occur
in that area.


**Author contribution**
**Miguel González-Pleiter**: identified the research question, formulated the hypothesis,
developed the experimental design, planned the experiments, performed the
experiments in the field, performed the experiments in the laboratory, compiled the
data sets, analyzed the data, discussed the results, prepared graphical material, wrote
the paper (original draft) and provided financial support. **Gissell Lacerot**: identified the
research question, formulated the hypothesis, developed the experimental design,
planned the experiments, checked the field data, discussed the results, wrote the paper
(final version). **Carlos Edo**: performed the experiments in the laboratory, compiled the
data sets, analyzed the data, discussed the results, prepared graphical material and
review final manuscript. **Juan Pablo Lozoya**: developed the experimental design,
checked the field data, discussed the results, review final manuscript and provided
financial support. **Francisco Leganés:** discussed the results, review final manuscript and
provided financial support. **Francisca Fernández-Piñas**: checked the field data, checked
the laboratory data, discussed the results, review final manuscript and provided
financial support. **Roberto Rosal**: checked the field data, checked the laboratory data,
discussed the results, review final manuscript and provided financial support. **Franco**
**Teixeira de Mello**: identified the research question, formulated the hypothesis,

developed the experimental design, planned the experiments, performed the experiments in the field, checked the field data, prepared graphical material, discussed the results, review final manuscript and provided financial support.

**Acknowledgements**

This research was funded by the Government of Spain (CTM2016-74927-C2-1/2-R) and the Uruguayan Antarctic Institute. MGP thanks the Carolina Foundation for the award of a postdoctoral grant (SEGIB). CE thanks the Spanish Government for the award of a predoctoral grant. The authors gratefully acknowledge the support of Fiorella Bresesti, Evelyn Krojmal and Barbara De Feo from the Centro Universitario Regional del Este, Universidad de la República for their assistance during sampling, of Marta Elena González Mosquera from University of Alcala for providing access to the Agilent Cary 630 FTIR spectrometer, and of Gastón Manta from Facultad de Ciencias, Universidad de la República for providing historical wind analysis at the Artigas Antarctic Research Base. FTM, GL and JPL thanks the Sistema Nacional de Investigadores (SNI) and the Programa de Desarrollo de las Ciencias Básicas (PEDECIBA).

**Declaration of competing interest**

The authors declare no conflict of interest.

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

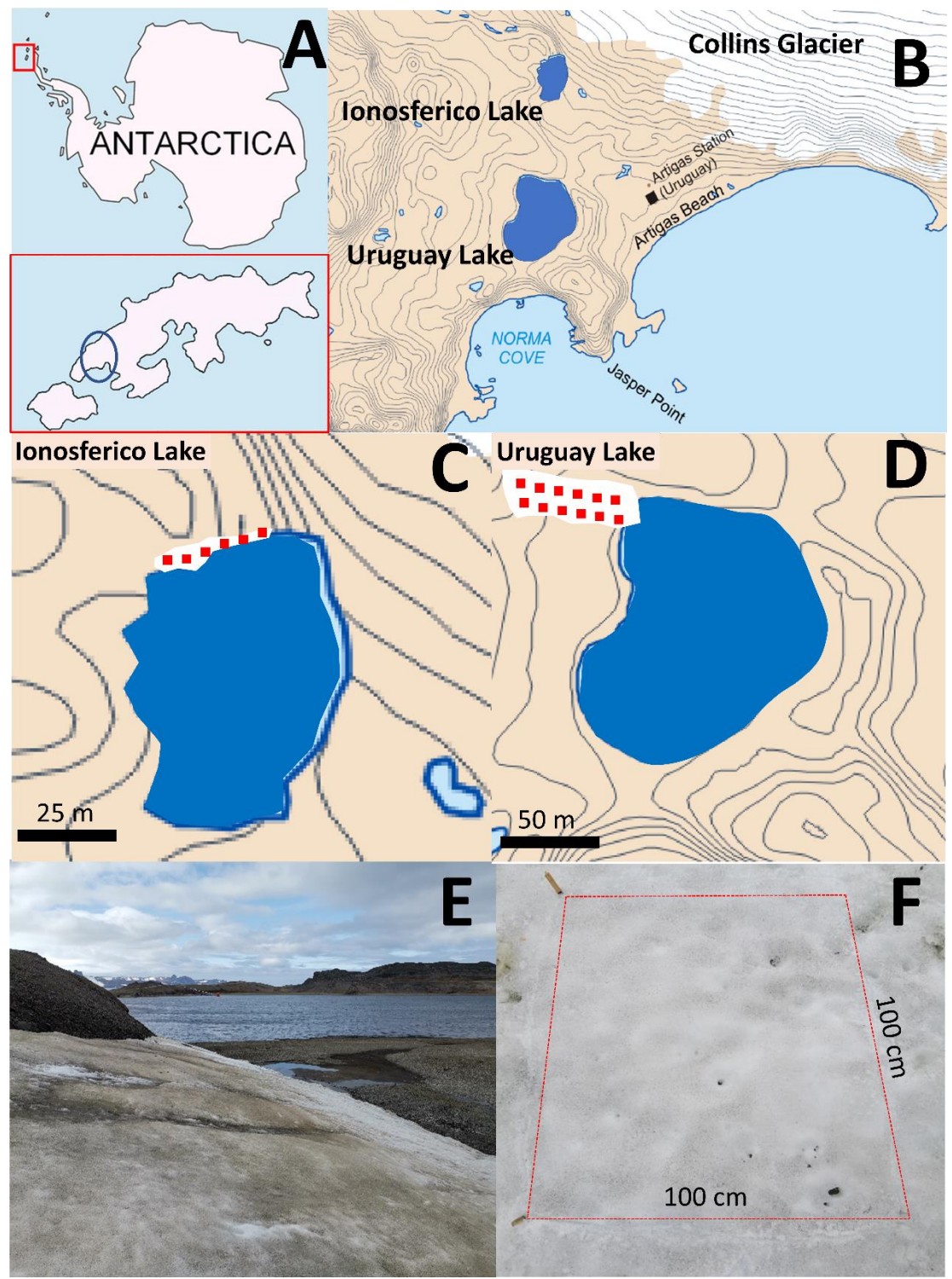

**Figure 1.** (A) General view of Antarctica and location of King George Island.  The blue
circle indicates the Fildes Peninsula. Collins Glacier is located on the northeast of Fildes
Peninsula. (B) A detailed view of Ionosferico lake, Uruguay lake, Artigas Research Station
and Collins Glaciers in the Fildes Peninsula. (C) and (D) ablation zone of Collins Glacier
close to Ionosferico lake and Uruguay lake, respectively. (E) Photograph of the glacier
surface close to Uruguay lake that constitute part of the ablation zone of Collins Glacier
taken on 18/02/2020. (F) A representative square on the glacier surface used in this
study.

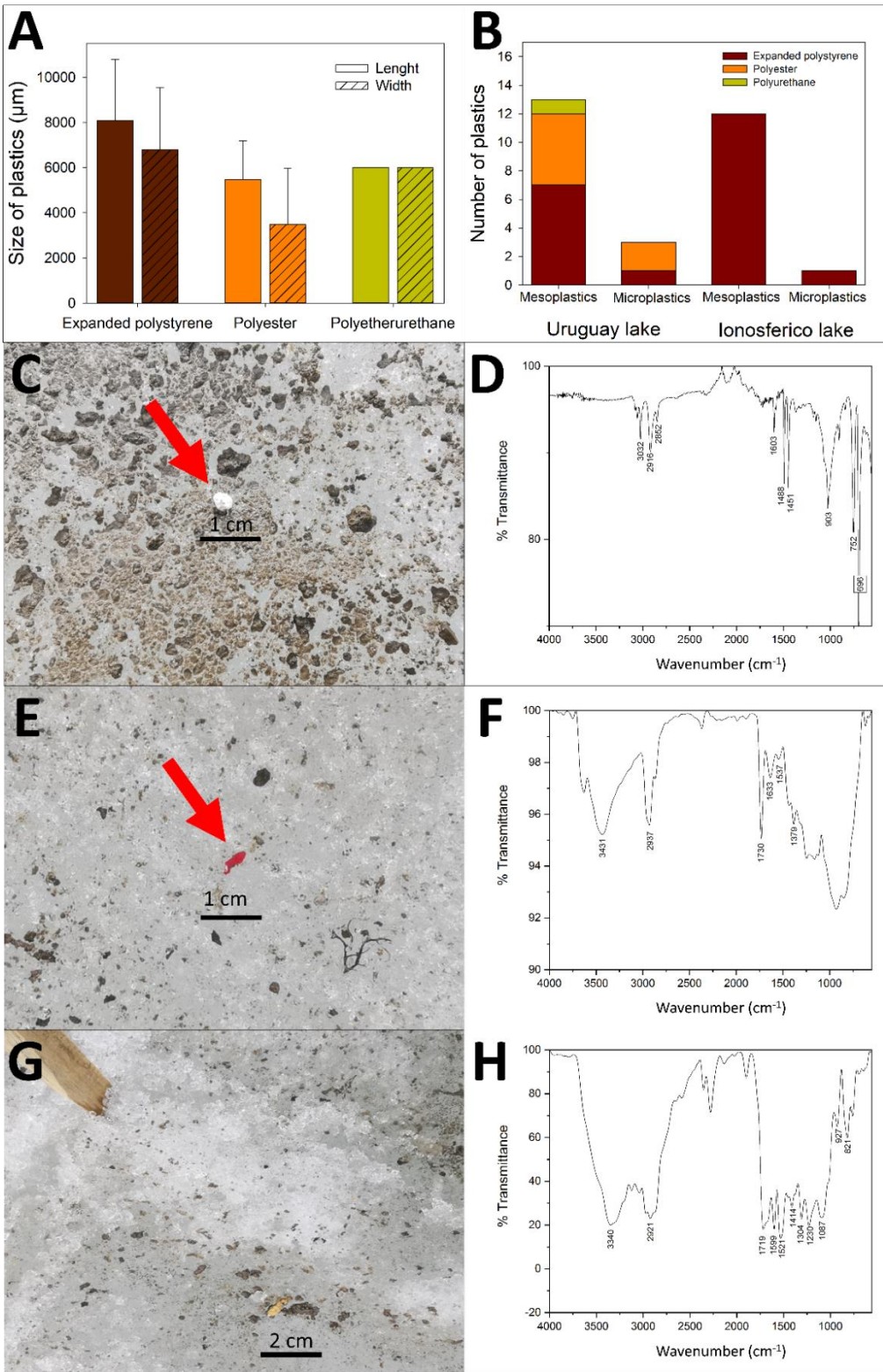

**Figure 2**. (A) Size of the plastics collected on the glacier surface. (B) Total number of the mesoplastics and microplastics found on the glacier surface close to Uruguay lake and Ionosferico. Representative photographs of expanded polystyrene (C), polyester (E) and polyurethane (G) found on the glacier surface. The red arrows indicate the plastics. FTIR representative spectra of expanded polystyrene (D), polyester (F) and polyurethane (H) found on the glacier surface.

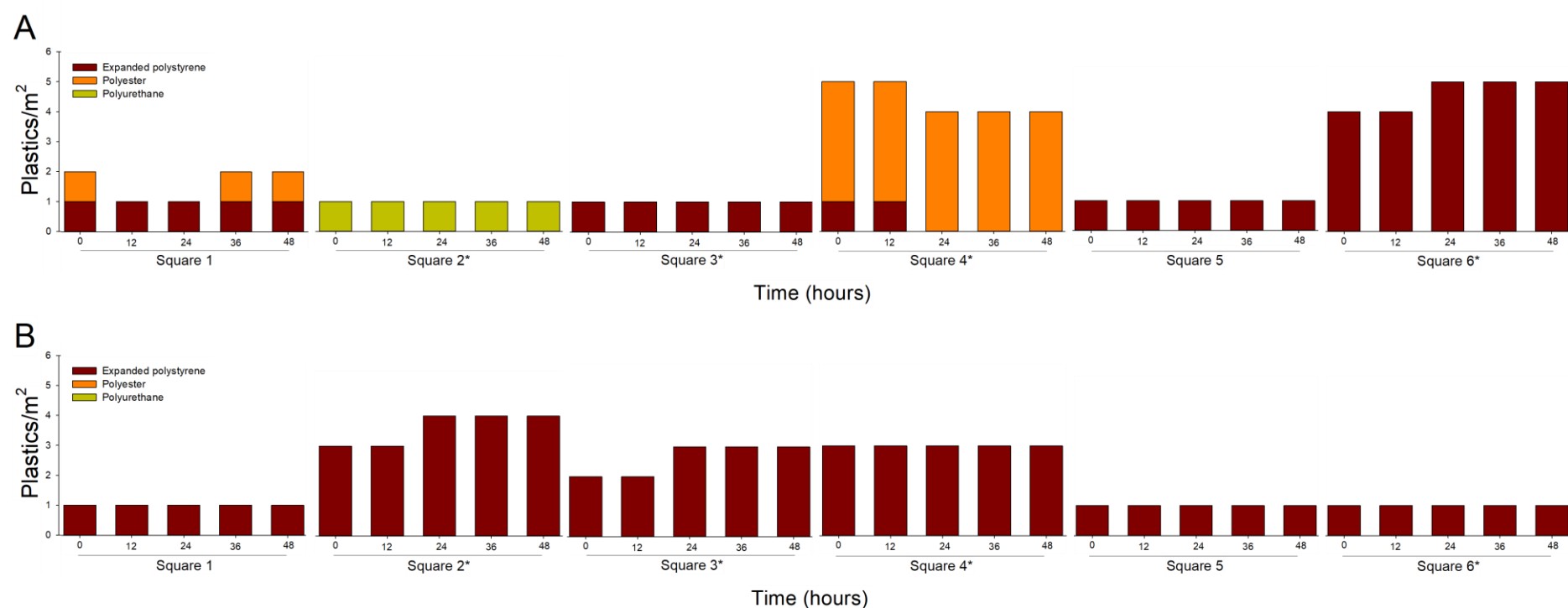

**Figure 3**. Changes in the presence of plastics into the squares marked on ice surface close to Uruguay lake (A) and close to Ionosferico lake (B)
that constitute part of the ablation zone of Collins Glacier in Maxwell Bay in King George Island (Antarctica). Plastics were monitored every 12 hours
for two days (18/2/2020 and 20/2/2020) in the absence of rainfall. Asterisks indicate squares different from those used to the assessment of
plastic concentration.

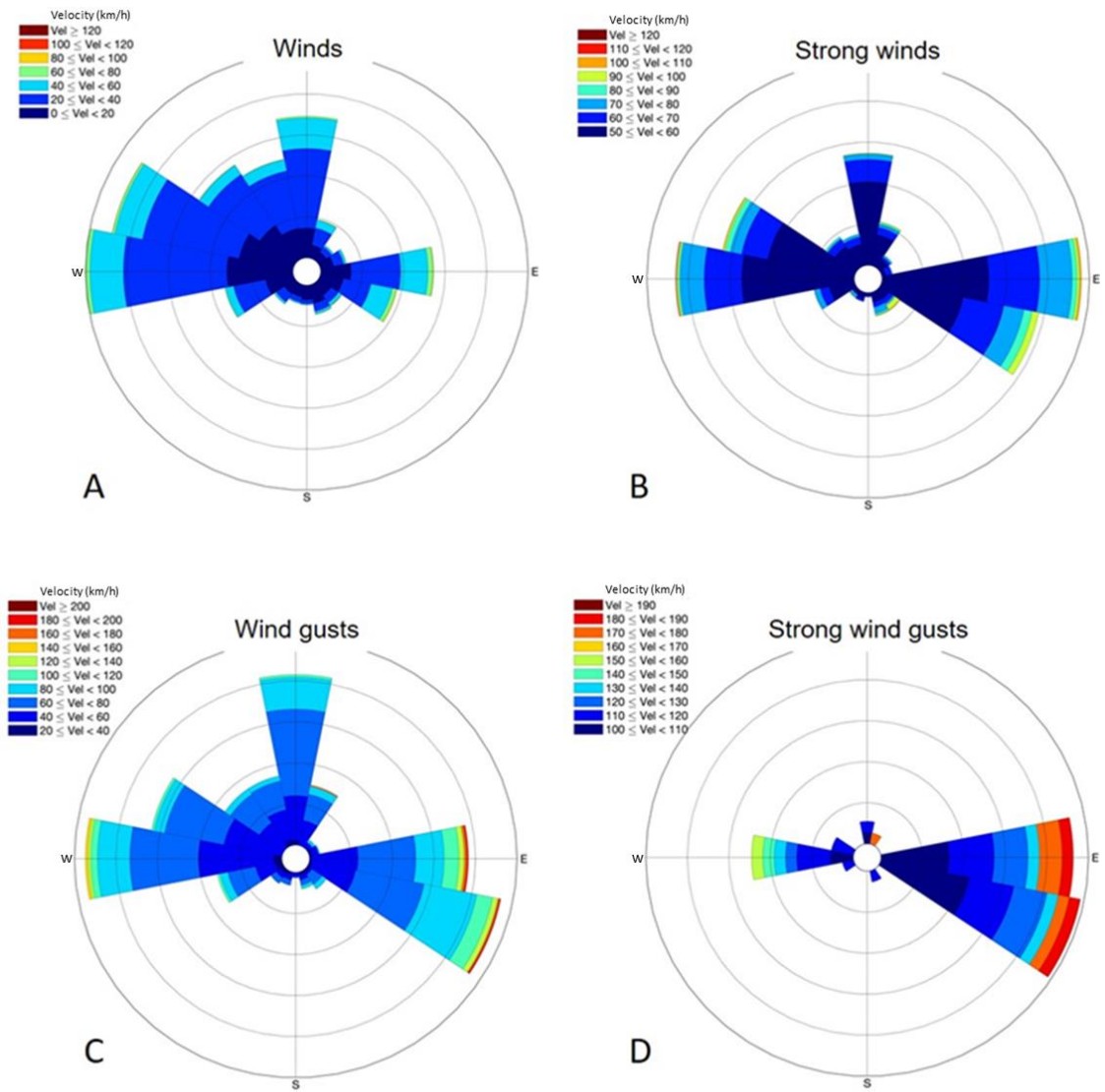

**Figure 4.** Wind Roses obtained for the area of BCAA based on historical data of the
Uruguayan National Institute of Meteorology (January 1998 - May 2016; 24,698
records). Based on the speed of winds considered (A) and (B) refer to *Winds* and *Strong*
*winds*, and (C) and (D) to *Wind Gusts* and *Strong wind gusts*, respectively.