# Peer review of "A pilot study about microplastics and mesoplastics in an Antarctic glacier: 1 the role of aeolian transport 2 3 Miguel González-Pleiter1,2+, Gissell Lacerot3, Carlos Edo1, Juan Pablo-Lozoya4, Francisco 4 5 Leganés2, Francisca Fernández-Piñas<sup"

_The Cryosphere, 2020_

## Referee Comment (RC1) · Anonymous Referee #1 · 2 Dec 2020

This is a short and interesting paper showing the first evidence of micro- and meso-plastic contamination on an Antarctic glacier. Methods are simple, and sample size is limited, however they are enough for a very first report on plastic occurrence in an Antarctic glacier.

---

## Referee Comment (RC2) · Anonymous Referee #1 · 2 Dec 2020

I have two minor remarks on the text: Line 49: I suggest "Despite its rate of ice loss has increased ..." Lines 51-52: I suggest "the Antarctic crysphere ... (Dirscherl et al 2020), its cap of ice covering up to ..."

---

## Referee Comment (RC3) · Rachel Obbard (Referee) · 8 Dec 2020

This is an interesting finding and will be a valid addition to the cannon of literature on mesoplastic presence in polar regimes once some improvements are made. There are some places where the English needs improvement, but first allow me to discuss a major issue and two minor ones affecting the scientific quality.

First, the major one. Their conclusion seems premature, given that the authors don't discuss local meteorological conditions, starting with as prevailing wind direction or

speed. If Figure 1 A and B have not been rotated (compass points on the inset would be useful!), prevailing wind would be from west to east, i.e. the lake area toward Artigas Beach. This makes their conclusion about the source suspect. They might be able to explain it based on weather during the collection period, i.e. recent high wind events. However, the alternative conclusion is that the particles were transported down off the glacier (unlikely unless research was being done there during this period), or from ships on the far side. Given their size, they probably aren't from longer range in the atmosphere, but the authors need to explain why this couldn't be the case. The authors could do this by discussing the wind speeds that would be needed to loft transport particles of this size (there is literature on this). Although this is a short communication, the authors need to examine these alternate explanations before they can conclude as they do.

In the paragraph Lines 121-131, the authors should discuss how they ensured that they themselves weren't the source of the particles they found. Could they have come off your clothes, your tools, your transport? Papers about micro (and meso) plastics generally address such potential contamination issues.

Line 145: Were you able to identify the particles which were not plastics? Were they soot, for instance? This might tell you something about the source of the plastics. For that matter, what are the activities at the Artigas Beach location? Is there incineration going on? Paint scraping? It would be interesting to know and might support your analysis.

The following are a few minor issues with English (and punctuation) by line number. Some affect understanding so should be corrected:

Lines 49-52: This sentence doesn't make sense as written. If you intend it as support for the preceding sentence, the first clause ("Despite . . . (Rignot et al., 2019)") doesn't seem to add anything. Perhaps you need to break this into two different supporting sentences.

Lines 52, 54, and a few others: You are missing the period after al in several places. Your citations should be of the form, Jones et al., <year>.

Line 57: Compartments isn't quite the right word here. How about calling them "discrete parts of the cryosphere"?

Line 68: There should be a comma after (79 %)

Line 208: "being EPS ubiquitous on the ice" doesn't make sense as written.

Thank you for the opportunity to review your paper.

---

## Referee Comment (RC4) · Melanie Bergmann (Referee) · 10 Dec 2020

1. General points: I recommend publication of this interesting MS in the journal Cryosphere since it is based on rare data from a remote area (Antarctica). In addition, it covers atmospheric microplastic pollution, an emerging issue that we do not know so much about until now. I also like the fact that larger plastic, mesoplastic, is addressed, for which we don't have so much data, especially in terms of atmospheric MP pollution. However, there are a few concerns that have to be tackled before this can be published in my opinion. Since you deal with microplastic, please provide a few lines on how you dealt with contamination (e.g. from yourselves). Did you do take a control? If you did not, this should at least be critically be discussed. The standard in microplastic research is nowadays to take blanks and give details on contamination prevention. Both of which is currently not mentioned at all. Based on this many reviewers would (rightly) reject the MS. However, since it is from such a remote place and deals with large micro- and mesoplastic, I think it merits publication. Still, this drawback needs to be communicated clearly. Please, make goals of this research clearer by formulating hypotheses or research questions: why you studied the two lake glaciers (e.g. test effect of distance to human plastic source). In addition, I strongly recommend that you then test your hypothesis on the data that you collected in both the sampling and experiment via statistical testing of abundance and polymer composition. Please add another graph, which shows the temporal trend over the 48-h experiment. Now this is hardly mentioned at all in the MS, which is a shame given all the work done. Please, also describe the trend in a separate section in results and consider statistical testing. Please, present all data as per m2, refrain from using numbers without a unit, which are meaningless.

2. Specific points Abstract -L. 31: Some readers may only be able to read the abstract. Therefore, please present the most important data here: the range of mean plastic pollution as well as microplastic that you recorded at the two sites in m-2, that polystyrene was the main polymer type found and also present the size range.

Introduction -L. 60: "The concentration of MP in snow is generally higher (0 to 1.5 x 105 MP L-1 of melted snow) near urban areas (Bergmann et al., 2019), than in sea ice (up to 12000 MP L-1 of melted ice), although there are large differences between studies even from the same region (Peeken et al., 2018; Von Friesen et al., 2020)." This is not a good comparison if you want to infer that plastic concentrations are higher in snow vs sea ice, as you compare snow concentrations from an urban (Bavarian Alps) area with sea ice concentrations from a remote area (Arctic). Please use the Arctic snow concentrations given in the paper, not the Bavarian ones! If you do so, the concentrations are similar. In addition, snow and sea ice are not so comparable as sea ice concentrates (microplastic) particles during ice formation. -L.65: Should it not rather be 'ice weight'? Because we are not really talking about sediments here but of ice?! -L.73: Other studies very likely found more small MP because they used methodology that can detect smaller particle sizes (e.g. u-FTIR imaging), this should be mentioned. Otherwise, it sounds as if MP in the Central Arctic were of larger size, which is likely only a detection bias. Please, add sth. like "due to the analytical methods used, which can capture smaller- sized plastic" -L74: "In general, the presence of plastics > 5mm are not reported in compartments of the cryosphere, probably due to the difficulty of large plastic items to reach the remote areas where these are located. I do not think that this is assumption is likely. Why should larger plastic not make it to the same places in the ocean as microplastic? I think the reason why they have not been reported in the marine cryosphere is that their concentrations are lower compared with MP. The likelihood of catching them by sampling gear is therefore also smaller, especially when analysing only small sample volumes. I would add sth. like: "In general, the presence of plastics > 5mm are not reported in compartments of the cryosphere, probably because they occur at lower concentrations and therefore often evade our detection." -L. 96: Please rephrase the strange term 'occurrence dynamics' -Please, add research questions/hypotheses in the end of the Introduction. Why the two lakes? Why did you do the experiment? Material & methods: I am missing any mention of fibres. Many (atmospheric) studies find that they predominate (possibly contamination). You do not refer to them at all except from the introduction. Did you exclude them (describe this)? If you did not find any, please also discuss this. Absher et al. reported fibres from Antarctic waters (Absher, T.M., Ferreira, S.L., Kern, Y., Ferreira, A.L., Christo, S.W., Ando, R.A., 2019. Incidence and identification of microfibers in ocean waters in Admiralty Bay, Antarctica. Environmental Science and Pollution Research 26, 292-298.)

-Please, provide distance (m) of each lake to the research station. -L.104: Is this the correct journal format for positions? If not please convert as appropriate with °N/°W etc.
-Please, add a paragraph on data analysis (s. below).

-Throughout the MS, please structure into 1. Assessment of glacial plastic pollution 2. Experimental assessment of atmospheric plastic deposition

Results/Discussion

-L.145-157: It is not clear which particles these numbers refer to. To the time zero (before experiment started)? Or do the figure refer to all items recorded? Please make this clear (s. above re structure). Also, please refer to number of items per m2 rather than presenting (meaningless) numbers throughout the text/tables/Figures.

-Please, add a headline referring to before Experiment and Dry-deposition experiment or similar, which makes it easier for the reader to follow (s. above).

-Please, present both total plastic abundance and microplastic abundance (in m-2) only, because other MP researchers may only want to compare their microplastic data with yours.

-L. 182: please add figures from your own research (mean or range in m-2) and provide figures from the papers that you cite for comparison.

-L. 184: How does this support " the notion that freshwaters could play a role in the life cycle of plastics in this region"? I do not understand? I don't think this conclusion can be drawn from the data, especially when consensus is building that the atmosphere is an important pathway/source of microplastic not vice versa? It could be concluded that it comes from the same source. However, you state elsewhere that your pollution may actually come from the research base, so this would be quite a different source compared to that of plastic in the ocean Please delete this or rephrase.

-L. 187: You can only make this statement if you have tested this. Because by looking at Fig. 2 B I could not be sure if there is actually a significant difference in the abundance of plastics at the two sites. But it would be interesting if you could: So, I suggest, you do some t-testing or similar (see also below).

[Figure]

-L.190: Please honor this statement of growing tourism with a citation.

-L. 193: please provide some examples/citations of long-range transport of small particles (e.g. dust, pollen, algae) by winds.

-In addition, it needs to be stated that when comparing your data with those of others: you looked employed quite a different methodology, which is better suited at capturing larger items compared with the many microplastic studies and that the data are not comparable strictly speaking. By saying so, I do not mean that you should not compare. It just needs to be mentioned.

-L.195: These are good statements but please beef them up with your own findings and those of other workers. Please also refer to other papers, which deal with research-based plastic pollution in Antarctica, to strengthen your line of argument. You could also argue that more needs to be done on microplastic pollution, given that it seems widespread in the southern Ocean. Along the lines of:

"Our research indicates that our research in sensitive remote areas such as Antarctica leaves a footprint, namely plastic pollution. While reports of research-based litter pollution on the seafloor and beaches date back as early as the 1970's (Dayton & Robillard 1971; Lenihan et al., 1990; Sander et al. 2009) the handling of waste has improved through the Antarctic Treaty System , Annex III 'Waste Disposal and Waste Management'. It requires treaty states to remove all plastic from Antarctica, with the only exception 100 being those plastics that can be incinerated without producing harmful emissions 101 (Antarctic Treaty Secretariat, 1998). However, once plastics are broken down into small 102 fractions and dispersed throughout the continent and nearby waters, management 103 measures become very difficult to address, as indicated by our data. Sander et al. (2009) also report ongoing pollution from research debris, which had not been removed. A more rigorous management of macro- and microplastics is therefore essential for preserving the integrity of sensitive polar environments."

In the Arctic, I also found evidence of increasing research vessel activities correlated to

increasing litter on the seafloor (Bergmann, M., Klages, M., 2012. Increase of litter at the Arctic deep-sea observatory HAUSGARTEN. Marine Pollution Bulletin 64, 2734–2741)." e.g. Lenihan, H.S., Oliver, J.S., Oakden, J.M., Stephenson, M.D., 1990. Intense and localized benthic marine pollution around McMurdo Station, Antarctica. Marine Pollution Bulletin 21, 422-430.

-In the discussion, you could discuss that Polystyrene particles, which you detected, have been shown to affect Antarctic biota, e.g. sea urchin Sterechinus neumayeri (Bergami, E., Krupinski Emerenciano, A., González-Aravena, M., Cárdenas, C.A., Hernández, P., Silva, J.R.M.C., Corsi, I., 2019. Polystyrene nanoparticles affect the innate immune system of the Antarctic sea urchin Sterechinus neumayeri. Polar Biology 42, 743-757.). Nanoplastic also affect Antarctic krill: Bergami, E., Manno, C., Cappello, S., Vannuccini, M.L., Corsi, I., 2020. Nanoplastics affect moulting and faecal pellet sinking in Antarctic krill (Euphausia superba) juveniles. Environment International 143, 105999.

Figure 2 / Data analysis

-Please indicate if the data presented are from the Time-0 or the deposition experiment. 2A: I would be more interested to see the size frequency of all microplastics, not necessarily ordered by polymer type or width/length. I would either rely on the widest dimension only (as do many MP researchers) or calculate the surface area of particles (width x length = area) and present a size frequency of this surface area. The first suggestion may produce data that are more comparable with those of other researchers, though. You can still provide a supplement where length and width are presented. In addition, you could test for significant differences between sizes of particles from the two lakes. Maybe the more distant lake harbours smaller particles as the larger items are not transported over longer distances?

-2B: You should present the means per m2 and give error bars in B as well, pertaining to Stdev, SEM or another metric of data dispersion. These numbers are meaningless without an area reference. You have 6 replicates if I did not misread the methods, so error bars should be possible. Please present all numbers in the whole text as means plus/minus SEM or similar in m-2.

-In addition, I strongly suggest carrying out t-test or similar on the number of microplastics recorded from the two lakes to test your assumption that the closer proximity to the research base caused a greater MP load on one lake. The abundances look similar to me, actually, as is. You could also test for significant differences in the polymer composition between the two lakes using multivariate data analyses such as PERMANOVA or ANOSIM (e.g. in PRIMER-e or r)

-Please do the same test for the experiment. Also, please provide a graph to show the temporal trend in mean plastic abundance per m2 over the 48-h period in addition to data given the table (which could go into the supplement)

Table 1 -Please provide only data on confirmed plastic in this table and get rid of the column 'Total Plastics confirmed by FTIR', otherwise it is confusing to the reader. You can show other particles in a supplementary. -How come some abundances are negative or positive?! -Please define: what is 'U' or replace by Plot 1, 2,. . ., 6,. . ., 12 -Please convert and present all data as per m2 or write in legend that data are presented as m-2. -Underneath the polymer types add a row with 'Total plastic'

I recommend publication of this interesting MS in the journal Cryosphere since it is based on rare data from a remote area (Antarctica). In addition, it covers atmospheric microplastic pollution, an emerging issue that we do not know so much about until now. I also like the fact that larger plastic, mesoplastic, is addressed , for which we don't have so much data, especially in terms of atmospheric MP pollution.

However, there are a few concerns that have to be tackled before this can be published in my opinion. Since you deal with microplastic, please provide a few lines on how you dealt with contamination (e.g. from yourselves). Did you take a control? If you did not, this should at least be critically be discussed. The standard in microplastic research is nowadays to take blanks and give details on contamination prevention. Both of which is currently not mentioned at all. Based on this many reviewers would (rightly) reject the MS. However, since it is from such a remote place and deals with large micro- and **mesoplastic**, I think it merits publication. Still, this drawback needs to be communicated clearly.

Please, make goals of this research clearer by formulating hypotheses or research questions: why you studied the two lake glaciers (e.g. test effect of distance to human plastic source). In addition, I strongly recommend that you then test your hypothesis on the data that you collected in both the sampling and experiment via statistical testing of abundance and polymer composition.

Please add another graph, which shows the temporal trend over the 48-h experiment. Now this is hardly mentioned at all in the MS, which is a shame given all the work done. Please, also describe the trend in a separate section in results and consider statistical testing.

Please, present all data as per m2, refrain from using numbers without a unit, which are meaningless.

**2. Specific points**

**Abstract**

-L. 31: Some readers may only be able to read the abstract. Therefore, please present the most important data here: the range of mean plastic pollution as well as microplastic that you recorded at the two sites in m-2, that polystyrene was the main polymer type found and also present the size range.

**Introduction**

-L. 60: "The concentration of MP in snow is generally higher (0 to 1.5 x 105 MP L-1 of melted snow) near urban areas (Bergmann et al., 2019), than in sea ice (up to 12000 MP L-1 of melted ice), although there are large differences between studies even from the same region (Peeken et al., 2018; Von Friesen et al., 2020)."

This is not a good comparison if you want to infer that plastic concentrations are higher in snow vs sea ice, as you compare snow concentrations from an urban (Bavarian Alps) area with sea ice concentrations from a remote area (Arctic). Please use the Arctic snow concentrations given in the

**Fig. 1.** Referee's report

[Figure]

**Fig. 2.** Annotated changes in provided document

**Supplement:**

[Figure]

Brief communication:

Atmospheric dry deposition of microplastics and mesoplastics in an

Antarctic glacier: The case of the expanded polystyrene.

Miguel González-Pleiter[1,2]†, Gissell Lacerot[3], Carlos Edo[1], Juan Pablo-Lozoya[4], Francisco

Leganés[2], Francisca Fernández-Piñas[2], Roberto Rosal[1], Franco Teixeira-de-Mello[5]†

[1]Department of Analytical Chemistry, Physical Chemistry and Chemical Engineering,

University of Alcala, Alcalá de Henares, E-28871 Madrid, Spain

[2]Departament of Biology, Faculty of Sciences, Universidad Autónoma de Madrid,

Cantoblanco, E-28049 Madrid, Spain

[3]Ecología Funcional de Sistemas Acuáticos, Centro Universitario Regional del Este,

Universidad de la República, Ruta nacional Nº9 y ruta Nº15, 27000 Rocha, Uruguay

[4]Centro Interdisciplinario de Manejo Costero Integrado del Cono Sur (C-MCISur), CURE

(UDELAR), Tacuarembó entre Av. Artigas y Aparicio Saravia, 20000 Maldonado, Uruguay

[5]Departamento de Ecología Teórica y Aplicada, Centro Universitario Regional del Este (CURE, UDELAR), Tacuarembó entre Av. Artigas y Aparicio Saravia, 20000 Maldonado,

Uruguay

†Corresponding authors:

Miguel González-Pleiter, email: mig.gonzalez@uam.es

Franco Teixeira-de-Mello, email: frantei@fcien.edu.uy

**Abstract**

Plastics have been found in marine water and sediments, sea ice, marine invertebrates, and penguins in Antarctica. However, there is currently no evidence of their presence in Antarctic glaciers. Our pilot study investigated plastic occurrence on two ice surfaces that constitute part of
the ablation zone of Collins Glacier (King George Island, Antarctica).

Our results showed concentrations of expanded polystyrene (EPS) in the 0.17-0.33 items m$^{-2}$ range.
We registered an atmospheric dry deposition between 0.08 and 0.17 items m$^{-2}$ day$^{-1}$ (February
2019). This is the first report of plastic pollution in an Antarctic glacier, to which it was probably transported by wind, possibly from local research activities.

[Figure]

**Introduction**

The cryosphere is defined as the frozen hydrosphere of the Earth system that consists of areas in which the temperatures are below 0°C for at least part of the year (NOAA, 2019). The greatest proportion of the cryosphere in terms of volume is in Antarctica. Although its extent of ice has increased in the last decades (Rignot et al 2019), it is estimated that the Antarctic cryosphere holds around 90% of Earth's ice mass (Dirscherl et al 2020) covering its cap of ice up to 6% of the planet during the austral winter (Shepherd et al 2018).

Furthermore, Antarctic cryosphere represents the majority of the world's freshwater (Shepherd et al 2018) being, probably, the largest freshwater ecosystem in the planet.

Plastics, especially microplastics (plastics items < 5 mm; MP), have been detected in several compartments of the cryosphere including alpine glaciers (Ambrosini et al.,

2019; Cabrera et al., 2020; Materić et al., 2020), snow (Huntingdon et al. 2020; Bergmann et al., 2019; Österlund et al., 2019) and sea ice (Obbard et al., 2014; Peeken et al., 2018; Geilfus et al., 2019; Kelly et al., 2020; La Daana et al., 2020; Von

Friesen et al., 2020). The concentration of MP in Arctic snow is generally lower (0 to $14.4 \times 10^3$ MP

$L^{-1}$ of melted snow) (Bergmann et al., 2019), than in sea ice (up to

12,000 MP $L^{-1}$ of melted ice), although there are large differences between studies and sites even from the same region (Peeken et al., 2018; Von Friesen et al., 2020, Bergmann et al., 2019). The use of different units in reporting MPs concentration in alpine glaciers such as number of items per mass of sediment weight (78.3 ± 30.2 MPs $kg^{-1}$ of sediments; Ambrosini et al., 2019) and mass of MPs per volume (0 to 23.6 ± 3.0 ng of MPs $mL^{-1}$; Materić et al., 2020), makes comparisons between studies difficult. Regarding the shape of the MP found in the cryosphere, fibers seem to be dominant in alpine glaciers (65 %) and sea ice (79 %)

followed by fragments (Ambrosini et al., 2019; La Daana et al., 2020). Concerning the size of MP, La Daana et al. (2020) reported a broad size distribution in sea ice, with 67%

of MP in the 500-5000 μm range. Other studies found lower sizes, however, with significant amounts (around 90%) of MPs smaller than 100 μm in snow and sea ice (Bergmann et al., 2019; Peeken et al., 2018; Ambrosini et al., 2019; Kelly et al., 2020) due to the analytical methods used, which can capture smaller-sized plastic. In general, the presence of plastics > 5mm are not reported in compartments of the cryosphere, probably due to the difficulty of large plastic items to reach the remote areas where these are located. MP identification using micro-Fourier transform-infrared spectroscopy (μ-FTIR) revealed that polyethylene terephthalate (PET), polyamide (PA), polyester (PE), varnish (acrylates/polyurethane), nitrile rubber, ethylene-propylene- diene monomer (EPDM) rubber, polypropylene (PP), varnish, rayon and polyurethane (PU) are the most common types of MPs found (Obbard et al., 2014; Peeken et al., 2018; Ambrosini et al., 2019; Bergmann et al., 2019; Kelly et al., 2020; La Daana et al., 2020;

Materić et al., 2020) in cryogenic matrices. On the other hand, sources for these MP detected in the cryosphere remain poorly understood. It has been suggested that they could be transported by wind before being deposited by both wet and dry deposition in remote

**Kommentiert [MB1]:** Huntington, A., Corcoran, P.L., Jantunen, L., Thaysen, C., Bernstein, S., Stern, G.A., Rochman, C.M., 2020. A first assessment of microplastics and other anthropogenic particles in Hudson Bay and the surrounding eastern Canadian Arctic waters of Nunavut. FACETS 5, 432-454.

**Kommentiert [MB2]:** Should it not rather be ice weight? We are not really talking about sediments here?!

**Kommentiert [MB3]:** I do not think that this is assumption is likely. Why should larger plastic not make it to the same places in the ocean as microplastic? I think the reason why they have not been reported in the marine cryosphere is that their concentrations are lower compared with MP. The likelihood of catching them is smaller, especially when analysing small sample sizes.

2017;

[revised manuscript text omitted]

The presence of plastics has been reported in different places in Antarctica such as sea ice
(Kelly et al. 2020), sea surface (Suaria et al. 2020; Lacerda et al. 2019; Isobe et al. 2017; Cincinelli et
* * *
**Margin annotations:**

Kommentiert [MB6]: Please add a headline referring to before Experiment and Dry-deposition experiment or similar, which makes it easier for the reader to follow.

Kommentiert [MB7]: Please use full name rather than abbreviations, this eases reading.

Kommentiert [MB8]: Kelly, A., Lannuzel, D., Rodemann, T., Meiners, K.M., Auman, H.J., 2020. Microplastic contamination in east Antarctic sea ice. Marine Pollution Bulletin 154, 111130.

Kommentiert [MB9]: Suaria, G., Perold, V., Lee, J.R., Lebouard, F., Aliani, S., Ryan, P.G., 2020. Floating macro- and microplastics around the Southern Ocean: Results from the Antarctic Circumnavigation Expedition. Environment International 136, 105494.

Kommentiert [MB10]: Lacerda, A.L.d.F., Rodrigues, L.d.S., van Sebille, E., Rodrigues, F.L., Ribeiro, L., Secchi, E.R., Kessler, F., Proietti, M.C., 2019. Plastics in sea surface waters around the Antarctic Peninsula. Scientific Reports 9, 3977.

Kommentiert [MB11]: Isobe, A., Uchiyama-Matsumo...

[Figure]

al. 2017; Barnes et al. 2010), beaches (Sander et al. 2009; Convey et al. 2002) , marine zooplankton (Absher et al., 2019), seafloor (Cunningham et al. 2020; Munari et al., 2017; Reed et al., 2018; Lenihan et al. 1990; Dayton & Robillard 1971), marine benthic invertebrates (Sfriso et al., 2020), fish (Cannon et al. 2016) and penguins (Le Chen et al. 2020; Laganà et al. 2019; Bessa et al., 2019) as well as other sea birds (Ibanez et al., 2020; Fijn et al., 2012; Creet et al., 1994; van Franeker & Bell, 1988). However, there was only one study about the presence of plastics in the Antarctic cryosphere that

**Kommentiert [MB12]:** Barnes, D.K.A., Walters, A., Goncalves, L., 2010. Macroplastics at sea around Antarctica. Marine Environmental Research 70, 250-252.

**Kommentiert [MB13]:** Sander, M., Costa, E.S., Balbao, T.C., Carneiro, A.P.B., Santos, C.R., 2009. Debris recorded in ice free areas of an Antarctic Specially Managed Area (ASMA): Admiralty Bay, King Georgia Island, Antarctic Peninsula. Neotropical Biology and Conservation 4, 36-39.
Convey, P., Barnes, D., Morton, A., 2002. Debris accumulation on oceanic island shores of the Scotia Arc, Antarctica. Polar Biology 25, 612-617.

**Kommentiert [MB14]:** Cunningham, E.M., Ehlers, S.M., Dick, J.T.A., Sigwart, J.D., Linse, K., Dick, J.J., Kiriakoulakis, K., 2020. High Abundances of Microplastic Pollution in Deep-Sea Sediments: Evidence from Antarctica and the Southern Ocean. Environmental Science & Technology.
Lenihan, H.S., Oliver, J.S., Oakden, J.M., Stephenson, M.D., 1990. Intense and localized benthic marine pollution around McMurdo Station, Antarctica. Marine Pollution Bulletin 21, 422-430.
Dayton, P.K., Robilliard, G.A., 1971. Implications of pollution to the McMurdo Sound benthos. Antarctic Journal, 53-56.

**Kommentiert [MB15]:** Cannon, S.M.E., Lavers, J.L., Figueiredo, B., 2016. Plastic ingestion by fish in the Southern Hemisphere: A baseline study and review of methods. Marine Pollution Bulletin 107, 286-291.

**Kommentiert [MB16]:** Le Guen, C., Suaria, G., Sherley, R.B., Ryan, P.G., Aliani, S., Boehme, L., Brierley, A.S., 2020. Microplastic study reveals the presence of natural and synthetic fibres in the diet of King Penguins (Aptenodytes patagonicus) foraging from South Georgia. Environment International 134, 105303. [...]

**Kommentiert [MB17]:** Laganà, P., Caruso, G., Corsi, I., Bergami, E., Venuti, V., Majolino, D., La Ferla, R., Azzaro, M., Cappello, S., 2019. Do plastics serve as a possible vector for the spread of antibiotic resistance? First insights from bacteria associated to a polystyrene piece [...]

**Kommentiert [MB18]:** Ibañez, A.E., Morales, L.M., Torres, D.S., Borghello, P., Haidr, N.S., Montalti, D., 2020. Plastic ingestion risk is related to the anthropogenic activity and breeding stage in an Antarctic top predator seabird species. Marine Pollution Bulletin 157, 111351 [...]

[Figure]

was carried out in Antarctic sea ice (Kelly et al. 2020). Here, we provide the first report of plastics in the freshwater cryosphere of Antarctica, namely in Antarctic glaciers.

The concentration of plastics found on the surfaces of two freshwater glaciers that constitute part of the ablation zone of Collins Glacier in Maxwell Bay (## - ## items m-2) are similar to those found in nearby Antarctic marine environments (e.g. ## -## microplastics m-2) (Cincinelli et al., 2017; Munari et al.,

2017; Reed et al., 2018) supporting the notion that freshwaters could play a role in the life cycle of plastics in this region. In our study, wind was probably the transportation mode of plastics to the ice from the anthropogenic activities that occur around these lakes, and differences in the concentration of plastics (higher in Uruguay lake) a consequence of its proximity to these anthropogenic activities. Notably, EPS is widely used as insulation material of old buildings in the area, and alkyd resins find use as external coatings. Besides, a growing number of tourists, exerts increasing pressure on the area. The long-range transport of plastic by wind would be supported by studies evidencing the transport of soil and propagules of terrestrial and marine invertebrates and grasses, mosses and algae (Nkem et al., 2006).

Our research indicates that our research in sensitive remote areas such as Antarctica leaves a footprint, namely plastic pollution. While reports of research-based litter pollution on the seafloor and beaches date back as early as the 1970's (Dayton & Robillard 1971; Lenihan et al., 1990; Sander et al. 2009) the handling of waste has improved through the Antarctic Treaty System, Annex III 'Waste Disposal and Waste Management'. It requires treaty states to remove all plastic from Antarctica, with the only exception being those plastics that can be incinerated without producing harmful emissions (Antarctic Treaty Secretariat, 1998). However, once plastics are broken down into small fractions and dispersed throughout the continent and nearby waters, management measures become very difficult to address, as indicated by our data. Sander et al. (2009) also report ongoing pollution from research debris, which had not been removed. A more rigorous management of macro- and microplastics is therefore essential for preserving the integrity of sensitive polar environments.

**201 Conclusion**

This is the first report of the presence of both MeP and MP in an Antarctic glacier, which was probably transported to the sites by wind. In total, three types of plastics were found on two glacier surfaces that constitute part of the ablation zone of Collins Glacier (King George

Island, Antarctica) being EPS ubiquitous on the ice. Our study shows that the management of plastic contamination in Antarctica should focus strongly on the waste and microplastic generated by anthropogenic activities that occur in this place, including scientific research.

**209 Author contribution**

**Miguel González-Pleiter**: identified the research question, formulated the hypothesis,
* * *
presence of

**Kommentiert [MB19]:** How does this support " the notion that freshwaters could play a role in the life cycle of plastics in this region"? I do not understand? I don't think this conclusion can be drawn from the data, especially when consensus is building that the atmosphere is an important pathway/source of microplastic not vice versa? It could be concluded that it comes from the same source. However, you state that your pollution may actually come from the research base, so this would be quite a different source compared to that of plastic in the ocean Please delete this or rephrase.

**Kommentiert [MB20]:** You can only make this statement if you have tested this. Because by looking at Fig. 2 B I could not be sure if there is actually a significant difference. But it would be interesting if you could, so I suggest, you do some t-testing or similar.

**Kommentiert [MB21]:** Please support this statement of growing tourism with a citation

**Kommentiert [MB22]:** Dayton, P.K., Robilliard, G.A., 1971. Implications of pollution to the McMurdo Sound benthos. Antarctic Journal, 53-56.

Antarctic Treaty area. In fact

developed the experimental design, planned the experiments, performed the
experiments in the field, performed the experiments in the laboratory, compiled the
data sets, analyzed the data, discussed the results, prepared graphical material, wrote
the paper (original draft) and provided financial support. **Gissell Lacerot**: identified the
research question, formulated the hypothesis, developed the experimental design,
planned the experiments, checked the field data, discussed the results, wrote the paper
(final version). **Carlos Edo**: performed the experiments in the laboratory, compiled the

data sets, analyzed the data, discussed the results, prepared graphical material and
review final manuscript. **Juan Pablo Lozoya**: developed the experimental design,
checked the field data, discussed the results, review final manuscript and provided
financial support. **Francisco Leganés:** discussed the results, review final manuscript and
provided financial support. **Francisca Fernández-Piñas**: checked the field data, checked
the laboratory data, discussed the results, review final manuscript and provided
financial support. **Roberto Rosal**: checked the field data, checked the laboratory data,
**225** discussed the results, review final manuscript and provided financial support. **Franco**
**Teixeira de Mello**: identified the research question, formulated the hypothesis,
developed the experimental design, planned the experiments, performed the
experiments in the field, checked the field data, prepared graphical material and
provided financial support.

**232** **Acknowledgements**
This research was funded by the Government of Spain (CTM2016-74927-C2-1/2-R) and
the Uruguayan Antarctic Institute. MGP thanks the Carolina Foundation for the award
of a postdoctoral grant (SEGIB). CE thanks the Spanish Government for the award of a
predoctoral grant. The authors gratefully acknowledge the support of Fiorella Bresesti,
Evelyn Krojmal and Barbara De Feo from the Centro Universitario Regional del Este,
Universidad de la República for their assistance during sampling and of Marta Elena
González Mosquera from University of Alcala for providing access to the Agilent Cary
630 FTIR spectrometer. FTM, GL and JPL thanks the Sistema Nacional de Investigación
(SNI) and FTM and GL the Programa de Desarrollo de las Ciencias Básicas (PEDECÍBA).

**243** **Declaration of competing interest**
The authors declare no conflict of interest.

**246** **References**
Absher, T.M., Ferreira, S.L., Kern, Y., Ferreira, A.L., Christo, S.W., Ando, R.A., 2019.
Incidence and identification of microfibers in ocean waters in Admiralty Bay, Antarctica.
Environmental Science and Pollution Research 26, 292-298.
Allen, S., Allen, D., Moss, K., Le Roux, G., Phoenix, V. R., & Sonke, J. E., 2020. Examination
of the ocean as a source for atmospheric microplastics. PloS one, 15(5), e0232746.
Allen, S., Allen, D., Phoenix, V. R., Le Roux, G., Jiménez, P. D., Simonneau, A., Binet, S.,
Galop, D., 2019. Atmospheric transport and deposition of microplastics in a remote
mountain catchment. Nature Geoscience, 12(5), 339-344.
Ambrosini, R., Azzoni, R.S., Pittino, F., Diolaiuti, G., Franzetti, A., Parolini, M., 2019. First
evidence of microplastic contamination in the supraglacial debris of an alpine glacier.
Environmental Pollution 253, 297-301.
Antarctic Treaty Secretariat, 1998. Annex III to the Protocol on Environmental Protection
to the Antarctic Treaty. Waste disposal and waste management. Accessed at
http://www.ats.aq on 3 Aug 2020.
Barnes, D.K.A., Walters, A., Goncalves, L., 2010. Macroplastics at sea around
Antarctica. Marine Environmental Research 70, 250-252.
Bergmann, M., Wirzberger, V., Krumpen, T., Lorenz, C., Primpke, S., Tekman, M.B.,

[Figure]

[Figure]

Gerdts, G., 2017. High quantities of microplastic in Arctic deep-sea sediments from the
HAUSGARTEN observatory. Environmental Science & Technology 51, 11000-11010.

**Kommentiert [MB23]:** Please replace by this article, which deals with MP in snow, instead of deep-sea sediments.
Bergmann, M., Mützel, S., Primpke, S., Tekman, M.B., Trachsel, J., Gerdts, G., 2019. White and wonderful? Microplastics prevail in snow from the Alps to the Arctic. Science Advances 5, eaax1157.

[Figure]

[Figure]

Bessa, F., Ratcliffe, N., Otero, V., Sobral, P., Marques, J.C., Waluda, C.M., Trathan, P.N.,
Xavier, J.C., 2019. Microplastics in gentoo penguins from the Antarctic region. Scientific
reports 9, 1-7.

Cabrera, M., Valencia, B.G., Lucas-Solis, O., Calero, J.L., Maisincho, L., Conicelli, B., Massaine Moulatlet, G., and Capparelli, M.V. (2020). A new method for microplastic sampling and isolation in mountain glaciers: A case study of one antisanal glacier, Ecuadorian Andes. *Case Studies in Chemical and Environmental Engineering* 2, 100051. https://doi.org/10.1016/j.cscee.2020.100051

Convey, P., Barnes, D., Morton, A., 2002. Debris accumulation on oceanic island shores of the Scotia Arc, Antarctica. Polar Biology 25, 612-617.

Cunningham, E.M., Ehlers, S.M., Dick, J.T.A., Sigwart, J.D., Linse, K., Dick, J.J., Kiriakoulakis, K., 2020. High Abundances of Microplastic Pollution in Deep-Sea Sediments: Evidence from Antarctica and the Southern Ocean. Environmental Science & Technology.

Creet, S., Van Franeker, J.A., Van Spanje, T.M., Wolff, W.J., 1994. Diet of the Pintado Petrel Daption Capense at King George Island, Antarctica, 1990/91. Marine Ornithology 22, 221-229.Dayton, P.K., Robilliard, G.A., 1971. Implications of pollution to the McMurdo Sound benthos. Antarctic Journal, 53-56.

Dayton, P.K., Robilliard, G.A., 1971. Implications of pollution to the McMurdo Sound benthos. Antarctic Journal, 53-56.

[revised manuscript text omitted]

Ionosferico lake

| Squares | 1 I | 2 I | 3 I | 4 I | 5 I | 6 I | Plastics $m^{-2}$ | Total plastics confirmed by FTIR |
|---|---|---|---|---|---|---|---|---|
| EPS | 1 | 0 | 0 | 0 | 1 | 0 | 0.33 | 2 |
| Polyester | 0 | 0 | 0 | 0 | 0 | 0 | 0 | 0 |
| Polyetherurethane | 0 | 0 | 0 | 0 | 0 | 0 | 0 | 0 |
| Total Plastic | 1 | 0 | 0 | 0 | 1 | 0 | | |

**Atmospheric dry deposition experiment**

Uruguay lake

| Squares | 1U | | | | | 2U* | | | | | 3U* | | | | | 4U* | | | | | 5U | | | | | 6U* | | | | | Total plastics confirmed by FTIR | Plastics $m^{-2}\,day^{-1}$ |
|---|---|---|---|---|---|---|---|---|---|---|---|---|---|---|---|---|---|---|---|---|---|---|---|---|---|---|---|---|---|---|---|---|
| Time (h) | 0 | 12 | 24 | 36 | 48 | 0 | 12 | 24 | 36 | 48 | 0 | 12 | 24 | 36 | 48 | 0 | 12 | 24 | 36 | 48 | 0 | 12 | 24 | 36 | 48 | 0 | 12 | 24 | 36 | 48 | | |
| EPS | 1 | | | | | | | | | | 1 | | | | | 1 | | -1 | | | 1 | | | | | 4 | 1 | | | | 8 | 0.08 |
| Polyester | 1 | -1 | | +1 | | | | | | | | | | | | 4 | | | | | | | | | | | | | | | 5 | 0.08 |
| Polyetherurethane | | | | | | 1 | | | | | | | | | | | | | | | | | | | | | | | | | 1 | 0 |
| Total Plastic | 2 | 1 | 0 | 1 | 0 | 1 | 0 | 0 | 0 | 0 | 1 | 0 | 0 | 0 | 0 | 5 | 0 | 1 | 0 | 0 | 1 | 0 | 0 | 0 | 0 | 4 | 0 | 1 | 0 | 0 | 14 | |

Ionosferico lake

| Squares | 1 I | | | | | 2 I *w | | | | | 3 I * | | | | | 4 I * | | | | | 5 I * | | | | | 6 I * | | | | | Total plastics confirmed by FTIR | Plastics $m^{-2}\,day^{-1}$ |
|---|---|---|---|---|---|---|---|---|---|---|---|---|---|---|---|---|---|---|---|---|---|---|---|---|---|---|---|---|---|---|---|---|
| Time (h) | 0 | 12 | 24 | 36 | 48 | 0 | 12 | 24 | 36 | 48 | 0 | 12 | 24 | 36 | 48 | 0 | 12 | 24 | 36 | 48 | 0 | 12 | 24 | 36 | 48 | 0 | 12 | 24 | 36 | 48 | | |
| EPS | 1 | | | | | 3 | | 1 | | | 2 | | 1 | | | 3 | | | | | 1 | | | | | 1 | | | | | 13 | 0.17 |
| Polyester | | | | | | | | | | | | | | | | | | | | | | | | | | | | | | | 0 | 0 |
| Polyetherurethane | | | | | | | | | | | | | | | | | | | | | | | | | | | | | | | 0 | 0 |
| Total Plastic | 1 | 0 | 0 | 0 | 0 | 3 | 0 | 1 | 0 | 0 | 2 | 0 | 1 | 0 | 0 | 3 | 0 | 0 | 0 | 0 | 1 | 0 | 0 | 0 | 0 | 1 | 0 | 0 | 0 | 0 | 13 | |

---

## Author Comment (AC1) · 9 Feb 2021

Thank you very much for your comment. You are right, the goal of this manuscript was to report the first evidence of plastic contamination on an Antarctic glacier, in the format of a brief communication. We think our findings are a significant advance for scientific investigations about plastics since no such studies have been conducted in Antarctic glaciers, a remote, and supposedly pristine area. We used an adequate methodology according to our goal, and the current scientific literature. Of course, further research is necessary to elucidate the extent of this contamination in Antarctic glaciers (especially

of the small fraction that were probably overlooked in this study due to the methodology used), their sources, and their impacts. Before all this, the first step was to demonstrate that plastics are there and, in this study, we have been able to show it.
* * *

---

## Author Comment (AC2) · 9 Feb 2021

Thank you very much for your comments.

- Line 49, we have written: "Despite the increasing rate of ice loss during last decades [. . .]" instead of "Despite that its rate of ice has increased [. . .].

- Line 51 - 52, we have written: "[. . .] it has been estimated that the Antarctic cryosphere holds around 90% of Earth's ice mass [. . .]" instead of "[. . .] the Antarctic cryosphere. . . (Dirscherl et al., 2020) covering its cap of ice up to [. . .] "

---

## Author Comment (AC3) · 9 Feb 2021

Thank you very much for your comments. It is a pleasure to discuss these issues with you. Following your suggestions, we have revised and improved the English, and we have discussed the issues that are affecting the scientific quality of our study. As you know, brief communications have a maximum of 3 figures and/or tables, a maximum of 20 references, a maximum of 4 pages, and no supplementary material; therefore, the space was very limited. For this reason, we have changed the article category to a research article, and we have included additional information to further clarify your

comments.

Major issue: Firstly, we have added the compass points in Figure 1 as an essential element in any map. Furthermore, we have included more information about the local meteorological conditions including prevailing winds (Figure 4) and daily wind patterns during the experiment (Figure S1). As you mentioned, the predominant winds in that area are from west. Historical data of the Uruguayan National Institute of Meteorology from BCAA (January 1998 - May 2016; 24,698 records) confirm that (Figure 4A), but also show that the strongest winds (i.e. gusts and strong gusts >150km/h) are mainly from east - southeast with some events from west (Figure 4B, C and D).

Figure 4. Wind Roses obtained for the area of BCAA based on historical data of the Uruguayan National Institute of Meteorology (January 1998 - May 2016; 24,698 records). Based on the speed of winds considered (A) and (B) refer to Winds and Strong winds, and (C) and (D) to Wind Gusts and Strong wind gusts, respectively.

Considering the used experimental strategy (i.e., checking the presence and removing 'new' plastics on a daily basis), the presence of plastics should be more related to the wind regimes that occurred on the days the study was conducted. Based on the available information we had access (i.e. Villa de la Estrellas, Fildes Peninsula climatological information, which is located near the Artigas Beach as shown in Figure S2A), during the study period (18/02/2020 - 20/02/2020) the wind was from northeast (45°) rotating to south (180°), with a speed between 10 and 30 km/h (Figure 2A). These wind conditions seem to suggest a possible link between Artigas Beach activities (BCAA and, especially, tourism) and wind-mediated aerial deposition of plastics.

Figure S1. Weather conditions in Villa Las Estrellas, during February 2020 based on the available information we had access to (i.e. La Villa de la Estrellas, Fildes Peninsula climatological information) (A), green rectangle indicates the study period (18/02/2020 and 20/02/2020), and (B) the distance between Villa Las Estrellas and Artigas Beach.

In general, the presence of microplastics and ,especially, mesoplastics on the surface

of these Antarctic glacier could be explained by the prevailing winds (i.e., strong winds, wind gusts and strong wind gusts), which could transport plastics from the Artigas Beach to the ice around both Uruguay and Ionosferico lakes. Atmospheric dry deposition of plastics could be the result of daily wind patterns was from northeast (45°) rotating to south (180°) during the collection period, which could have also transported plastics from the Artigas Beach. Therefore, plastic wastes present on the Artiga Beach, which are probably released from marine environments, and the human activities (e.g.: tourisms) could be the source of the plastics reaching glacier and atmospheric dry deposition could play a key role in their transport. Regarding mesoplastics found, they had very low density (e.g. EPS and PU), which probably eased their transport.

Regarding the alternative explanation ("that the particles were transported from ships on the far side"), it does not seem too likely considering the wind direction on the days of the experiment (Figure S1A). Considering only the predominant winds this could be, but the strongest winds, which could move the mesoplastics long distances, come predominantly from the E-SE.

It should be notes that this is a first step which pretends to show that meso and microplastics are present in Antarctic glaciers, and undoubtedly further researches are necessary to elucidate their distribution, their sources, pathways and trajectories (e.g. using HYSPLIT, LAGRANTO, FLEXPART), and of course their possible impacts. Based on the wind information we were able to collect; we have modified our discussion as follows: "In this sense, winds (especially high-speed ones) appear to be a key element in the transport of plastics to Antarctic glaciers. The prevailing winds in the study area (Figure 1B) blow predominantly from the west (Figure 4A). However, strong winds (Figure 4B), wind gusts (Figure 4C), and strong wind gusts (Figure 4D) blow mainly from the east and southeast directions, and could be responsible for the spreading of plastics from the different origins to the surface of the glacier ablation areas. These strong winds would explain the presence of MePs despite their size (Figure 2A). In fact, the low density of the MePs found (mainly EPS; Figure 2B) would have allowed their easy

dispersion by wind.

Our results on the dry deposition of plastics support the hypothesis that the role of the wind is essential for the transport of MPs and MePs in (and among) different areas of Antarctica. The dry deposition of plastics (Table S2) was closely related to the wind regimes during the study period (Figure S1). Based on information available on the meteorological conditions during the study dates (18/02/2020 - 20/02/2020) in La Villa de la Estrellas (Figure S1A), which is located near the Artigas Beach (Figure S2B), the wind blew from the northeast veering to the south with a speed between 10 and 30 km/h (Figure S1A). These wind conditions suggest a possible link with marine environment, which can act as a source of plastics (Allen et al., 2020), and potentially explain the presence of plastics on the glacier ablation areas. However, considering the low intensity of the winds recorded during those days (Figure S1A) and the presence of MePs, it is also possible that the predominant high-speed winds transported MePs from other adjacent areas of the Fildes Peninsula to the vicinity of the lakes, in the days prior to our study (Figure 4B, C, and D) and then, the milder winds registered during the sampling days (Figure S1A) deposited these MePs on the ice."

Minor issues: Lines: 121 - 131. Regarding the sample contamination, all the materials used (metal, steel and glass) were previously cleaned with MilliQ water, wrapped in aluminum foil and heated up to 300 °C for 4 hours in order to remove all possible rests of organic matter. The use of any plastic material was avoided. Furthermore, possible contamination due to clothing was controlled throughout the whole process by comparing clothes fibres and fragments with our samples. Moreover, it should be noted that the types of plastics found in our study are not typically associated with clothing, or any of our sampling tools. In fact, some of them (e.g. EPS) are not even allowed currently in the scientific bases and were not part of any of our sampling gear. Given their size, plastics found in this study were detected by the naked eye and their traceability was easily maintained during quantification and identification of the samples. We have incorporated this in our manuscript, as follows (lines: 153 -161): "2.5 Prevention
of procedural contamination. To avoid sample contamination, all materials used were previously cleaned with MilliQ water, wrapped in aluminum foil, and heated to 300 °C for 4 h to remove organic matter. The use of any plastic material during sampling was avoided. Furthermore, possible contamination from our clothes was controlled throughout the sampling, by checking fibers and fragments extracted from the clothes against the MPs and MePs found in the samples, and by positioning us against the wind during sampling. Given their size, plastics found in this study were detected by the naked eye and their traceability could be easily maintained during quantification and identification of the samples."

Line 145: Regarding the identification of the particles, 16 items were not confirmed as plastic by FTIR or $\mu$FTIR analysis. These items were not considered plastic materials because they were not identified as a known material with matching values > 60%. Some of these spectra could show some similarities with alkyd resin (polyester modified by the addition of other components), which are widely used in many synthetic paints. However, none of them resembled soot. Regarding plastics identified, the types found (Figure 2B) are related to human activities carried out in the Artigas area. For instance, EPS are widely used in packaging and (together with the PU) as insulation material in old buildings in this area and alkyd resins found are used as external coatings. We have incorporated this in our manuscript, as follows (lines: 233 - 249): "The chemical composition of the plastics found (Figure 2D, F, and H) supports the fact that the source of the plastics could be of marine and/or land-based origin. The types of plastics found (Figure 2B) are related to human activities in the Fildes Peninsula that could generate plastic debris such as tourism, leaks in waste management at scientific bases or the presence of abandoned infrastructures. Considering the location of Collins Glacier and the main human activities on the Fildes Peninsula (e.g. airfield, scientific bases), the prevailing winds from the west could have transported small and lightweight plastics to the study area. In fact, EPS is widely used in packaging and as insulation material in old buildings in this area and polyester is also a component of old buildings paints. In the same way, some of these plastics could be released from

the marine environment to Artigas beach area and, then, be transported by the wind to the glaciers. In this sense, polyurethane MePs (which are similar to those found in this work) have already been reported in sea surface waters in the Antarctic (Jones-Williams et al., 2020) and EPS MePs have been found on Artigas beach (Laganà et al., 2019). These findings highlight a potential threat to the fragile Antarctic ecosystem, since the presence of these plastics (e.g. polystyrene particles) has been shown to affect Antarctic biota (Bergami et al., 2019; Bergami et al., 2020a)."

Lines 49-52, we have written: "Despite the increasing rate of ice loss during last decades (Rignot et al., 2019), it has been estimated that the Antarctic cryosphere holds around 90% of Earth's ice mass (Dirscherl et al., 2020)."

Line 52 and 54, we have revised and corrected all references.

Line 57. We have removed "compartments".

Line 68: We have included (line: 71) a comma after (79%)

Line 237: We have written: "EPs was ubiquitous in the two glacier surfaces studied"
* * *
[Figure]

[Figure]

**Fig. 1.**

[Figure]

Fig. 2.

---

## Author Comment (AC4) · 9 Feb 2021

Thank you very much for your comments, and the opportunity to discuss our study with you. As you know, brief communications have a maximum of 3 figures and/or tables, a maximum of 20 references, a maximum of 4 pages, and no supplementary material; therefore, the space is very limited. For this reason, we have changed the article category to a research article, and we have included additional information to further clarify your comments.

[Figure]

Regarding the sample contamination, all the materials used (metal, steel and glass) were previously cleaned with MilliQ water, wrapped in aluminum foil and heated up to 300 °C for 4 hours in order to remove all possible rests of organic matter. The use of any plastic material was avoided. Furthermore, possible contamination due to clothing was controlled throughout the whole process by comparing clothes fibres and fragments with our samples. Moreover, it should be noted that the types of plastics found in our study are not typically associated with clothing, or any of our sampling tools. In fact, some of them (e.g. EPS) are not even allowed in the scientific bases and were not part of any of our sampling gear. Given their size, plastics found in this study were detected by the naked eye and their traceability was easily maintained during quantification and identification of the samples. We have incorporated this in our manuscript, as follows (lines: 153 -161): "2.5 Prevention of procedural contamination. To avoid sample contamination, all materials used were previously cleaned with MilliQ water, wrapped in aluminum foil, and heated to 300 °C for 4 h to remove organic matter. The use of any plastic material during sampling was avoided. Furthermore, possible contamination from our clothes was controlled throughout the sampling, by checking fibers and fragments extracted from the clothes against the MPs and MePs found in the samples, and by positioning us against the wind during sampling. Given their size, plastics found in this study were detected by the naked eye and their traceability could be easily maintained during quantification and identification of the samples."

About the hypothesis of our research. Given the fact that plastics have already been found in other parts of the cryosphere (alpine glaciers, snow and sea ice) and in Antarctica (seawater, freshwater, sediments and organisms), our research question was: could plastic be found on Antarctic glaciers? and, does dry deposition (i.e. by wind) play a role in its transport from areas with human activities?. Following these research questions, the hypothesis in our original manuscript was "Our hypothesis is that plastics have also reached freshwater glaciers in Antarctica and that the dry deposition could be playing a crucial role in this process". To assess this, we chose two ice surfaces areas (an area around Uruguay lake and another around Ionosferico

lake) that constitute part of the ablation zone of Collins Glacier (King George Island, Antarctica). The reason for this choice is that we could easily access from BCAA to both areas on foot as often as the experiment required. Uruguay lake is located ∼300 m from Antarctic Scientific Base and Ionosferico lake is located ∼600 m from Artigas Base (see section 2.1 in material and methods). These relative differences in human pressure and distance from Artigas Beach could be evaluated in future studies to test the effect of distance to human plastic source in their atmospheric dry deposition of plastics in Antarctica. However, our goal in this pilot study was not to test this. In fact, plastics collected in our study are not enough to perform a robust statistical test in order to test this. Furthermore, we believe that other factors such as topography and a more detailed sampling gradient would have been necessary if that was our goal. Therefore, we have written: "So far, plastics have been found in specific parts of the cryosphere (mountain glacier, snow, and sea ice) and Antarctica (seawater, freshwater, sediments, and organisms). We hypothesize that plastics have also reached freshwater glaciers in Antarctica and that atmospheric dry deposition plays a crucial role in this process. To test this hypothesis, we carried out a pilot study to investigate the presence of plastics on two ice surfaces (one area close to Uruguay lake and another one close to Ionosferico lake) that constitute part of the ablation zone of Collins Glacier in Maxwell Bay in King George Island (Antarctica). Furthermore, the daily changes in the presence of plastics in these ice surfaces was evaluated in the absence of rainfall, to clarify the role of wind in their transport."

According your request, we have added another graph showing the temporal trend over the 48 hours in each squares of both ice surface (an area around Uruguay lake and another around Ionosferico lake) that constitute part of the ablation zone of Collins Glacier in Maxwell Bay in King George Island (Antarctica).

Figure 3. Changes in the presence of plastics into the squares marked on ice surface close to Uruguay lake (A) and close to Ionosferico lake (B) that constitute part of the ablation zone of Collins Glacier in Maxwell Bay in King George Island (Antarctica).

Plastics were monitored every 12 hours for two days (18/2/2020 and 20/2/2020) in the absence of rainfall. Asterisks indicate squares different from those used to the assessment of plastic concentration.

Following your request, we have structured the MS including in the results and material and methods new subtitles.

Besides, we agree with the reviewer about the importance of showing data as per m2. In this sense, all data have been also presented as plastics per m2 throughout the MS (. In fact, we have included two tables to clarify the results of both experiments (the assessment of plastic concentration and the assessment of atmospheric dry deposition of plastics). Furthermore, we considered relevant to include the total number of items identify as plastics with respect the total items collected as well as their characterization (see section 3.1 Characterization and identification of the plastics) in order to show the importance of the item identification using appropriate techniques (e.g. FTIR, RAMAN, HPLC-MS/MS).

Table S1. Characteristics of plastics found into the squares used for the assessment of plastic concentration on ice surface close to Uruguay lake and close to Ionosferico lake that constitute part of the ablation zone of Collins Glacier in Maxwell Bay in King George Island (Antarctica).

Area ID Square Polymer Color Size 1 ($\mu$m) Size 2 ($\mu$m) Type Uruguay 1 EPS White 4100 4022 Microplastic Uruguay 1 Polyester Red 4822 2544 Microplastic Uruguay 2 Polyester Red 6662 3747 Macroplastic Uruguay 3 not detected - - - - Uruguay 4 not detected - - - - Uruguay 5 EPS White 12628 11334 Macroplastic Uruguay 6 not detected - - - - Uruguay 7 not detected - - - - Uruguay 8 not detected - - - - Uruguay 9 not detected - - - - Uruguay 10 not detected - - - - Uruguay 11 not detected - - - - Uruguay 12 Polyester Red 2292 1356 Microplastic Uruguay Mean 0.17 EPS/m2 and 0.25 Polyester/m2 Ionosferico 1 EPS White 7583 5591 Macroplastic Ionosferico 2 not detected - - - - Ionosferico 3 not detected - - - - Ionosferico 4 not detected - - - - Ionosferico 5 EPS White 3817 3318 Microplastic Ionosferico 6 not detected - - - - Ionosferico Mean 0.33 EPS/m2

Table S2. Characteristics of plastics found at the end of the experiment into the squares used for the assessment of atmospheric dry deposition of plastics on ice surfaces that constitute part of the ablation zone of Collins Glacier (King George Island, Antarctica).

Area ID Square Polymer Color Size 1 ($\mu$m) Size 2 ($\mu$m) Type Uruguay 1 EPS White 4100 4022 Microplastic Uruguay 1 Polyester Red 4822 2544 Microplastic Uruguay 2* Polyurethane Brown > 5000 > 5000 Macroplastic Uruguay 3* EPS White 9301 8265 Macroplastic Uruguay 4* Polyester Red 6989 6834 Macroplastic Uruguay 4* Polyester Red 6168 5891 Macroplastic Uruguay 4* Polyester Red 5909 501 Macroplastic Uruguay 4* Polyester White > 5000 > 5000 Macroplastic Uruguay 5 EPS White 12628 11334 Macroplastic Uruguay 6* EPS White 9720 7963 Macroplastic Uruguay 6* EPS White 6292 5567 Macroplastic Uruguay 6* EPS White 9192 9023 Macroplastic Uruguay 6* EPS White 5595 4574 Macroplastic Uruguay 6* EPS White 7847 3640 Macroplastic Ionosferico 1 EPS White 7583 5591 Macroplastic Ionosferico 2* EPS White 6437 5220 Macroplastic Ionosferico 2* EPS White 10932 7572 Macroplastic Ionosferico 2* EPS White 5278 4726 Macroplastic Ionosferico 2* EPS White 9363 9186 Macroplastic Ionosferico 3* EPS White 9209 7932 Macroplastic Ionosferico 3* EPS White 7946 3834 Macroplastic Ionosferico 3* EPS White 13155 7925 Macroplastic Ionosferico 4* EPS White 7007 6905 Macroplastic Ionosferico 4* EPS White 7094 5112 Macroplastic Ionosferico 4* EPS White 16737 16085 Macroplastic Ionosferico 5 EPS White 3817 3318 Microplastic Ionosferico 6* EPS White 11576 11105 Macroplastic

Asterisks indicate squares different from those used for the assessment of plastic concentration.

It should be noted that in our new version of the manuscript we have also added more information and further discussed the role of wind intensity and direction in the area, in

order to give more insight into the possible influence of this environmental variable.

Specific points Line 31 (lines: 29 - 41): Thank you for your comment, we have included the most important data in the abstract: "Plastics have been found in several compartments in Antarctica. However, there is currently no evidence of their presence in Antarctic glaciers. Our pilot study investigated plastic occurrence on two ice surfaces (one area close to Uruguay lake and another one close to Ionosferico lake) that constitute part of the ablation zone of Collins Glacier (King George Island, Antarctica). Our results showed that expanded polystyrene (EPS) was ubiquitous ranging from 0.17 to 0.33 items m-2 whereas polyester was found only on the ice surface close to Uruguay lake (0.25 items m-2). Furthermore, we evaluated the daily changes in the presence of plastics in these areas in the absence of rainfall to clarify the role of the wind in their transport. We registered an atmospheric dry deposition rate between 0.08 items m-2 day-1 on the ice surface close to Uruguay lake and 0.17 items m-2 day-1 on the ice surface close to Ionosferico lake. Our pilot study is the first report of plastic pollution presence in an Antarctic glacier, possibly originated from local current and past activities, and the first to assess the effect of wind in its transport."

Line 60: We have written (lines: 60 – 65): "The occurrence of MPs in snow ranged from 0 to 1.5 x 105 MP L-1 of melted snow (Bergmann et al., 2019), although it should be noted that a part of this study was conducted near urban areas. Regarding sea ice, concentrations of up to 1.2 x 104 MP L-1 have been reported, although there are large differences between studies even from the same region (Peeken et al., 2018; von Friesen et al., 2020)."

Line 65: Ambrosini et al 2019 reported the occurrence of plastics as "items kg−1 of sediment (dry weight)". Checking section 2.2. Sample collection of their article we found the following description: "collected two cryoconite samples and four samples of sparse and fine (<2 mm) supraglacial debris from the ablation area of Forni Glacier". We have modified our manuscript to reflect this as: We have written (lines: 66 – 67): "[. . .] of ice weight (78.3 ± 30.2 MPs Kg-1 of sparse and fine supraglacial debris; Ambrosini et al., 2019) and mass [. . .]" instead of "[. . .] of sediment weight (78.3 ± 30.2 MPs Kg-1 of sediments; Ambrosini et al., 2019) and mass [. . .]"

Line 73: We have written: "The differences between these studies may be due to the different analytical methods used, particularly methodologies such as micro Fourier transform infrared spectroscopy ($\mu$FTIR, which can identify smaller sized MPs)."

Line 74: We have written (line 76 – 79): "In general, the presence of plastics > 5mm are not reported in discrete parts of the cryosphere, probably because they occur at lower concentrations and therefore often evade our detection" inside of "In general, the presence of plastics > 5mm are not reported in compartments of the cryosphere, probably due to the difficulty of large plastic items to reach the remote areas where these are located."

Line 96: Please, see lines: 94 -103.

In general, we excluded fibers from our study, since they were non detectable with the naked eye, and would have required ice extraction, melting and posterior water analysis, impacting our sampling strategy (i.e ice extraction from sampling squares). We have now added this information in the materials and methods section 2.2 (lines: 127 - 130): "It should be noted that our sampling strategy excluded the plastics non-detectable by the naked eye (i.e. small plastics such as fibers). Thus, we probably underestimated the concentration of small plastics on the ice surface."

Regarding the distance of each lake to the Artigas Scientific Base, we have now added this information in the new version of the manuscript (lines: 112 – 115).

Figure S. Distance between Artigas Scientific base and Ionosferico lake ~600 m (A). Distance between Artigas station and Uruguay lake ~300 m (B).

Line 104: Following your request, we have modified geographical positions as follows: "(S 62° 11' 6.54", O 58° 54' 42.23")" and "(62° 11' 59.41", O 58° 57' 44.17")"

According our request, we have added a paragraph on data analysis (see section 2.4)

Following your request, we have structured the MS including in the results and material and methods new subtitles.

Line 145-157: To clarify , we have added Figure 3 and a new Table S2 showing the temporal trend over the 48 hours in each squares of both ice surface (an area around Uruguay lake and another around Ionosferico lake) that constitute part of the ablation zone of Collins Glacier in Maxwell Bay in King George Island (Antarctica).

Lines 182: Following your request, we have compared our results in m-2 with the papers that we cited (lines: 251 -262)

Line 184: We have deleted this.

Line 187: These relative differences in human pressure and distance from Artigas Beach could be evaluated in future studies to test the effect of distance to human plastic source in their atmospheric dry deposition of plastics in Antarctica. However, our goal in this pilot study was not to test this. In fact, plastics collected in our study are not enough to perform a robust statistical test to test this.

Line 190: We have incorporated this in our manuscript (see discussion). Line 193: Here, we have mentioned: "Our results show that the atmospheric deposition of plastics on glaciers is very low being between two and four orders of magnitude lower than what is generally found in the rest of the continents (Dris et al. 2016; Cai et al 2017; Klein and Fischer, 2019; Brahney et al 2020). This could be due to the fact that we have used a different methodology that those used in previous studies and that probably underestimated the concentration of plastics, especially small fractions. Nevertheless, further research is necessary to elucidate the distribution, sources, pathways and trajectories, and impacts on this ecosystem of the plastics".

Line 195: We have modified our discussion

Figure 2 and Table 1: To clarify , we have added a new Figure 3 and a new Table S2 showing the temporal trend over the 48 hours in each squares of both ice surface (an

area around Uruguay lake and another around Ionosferico lake) that constitute part of the ablation zone of Collins Glacier in Max

[Figure]

[Figure]

[Figure]

**Fig. 1.**

[Figure]

[Figure]

**Fig. 2.**

---

## Author Response (AR2)

Re. GonzaleZ -Pleiter et al. A pilot study about microplastics and mesoplastics in an Antarctic glacier: the role of atmospheric dry deposition

This MS has already been thoroughly and ably reviewed by Dr. Berghmann and Dr. Obbard and I agree with their appraisals. Given the length of time this MS has already been in review, I don't want to delay decisions so I will give a brief review.

I am convinced the authors found plastic on the glacier, which is very interesting and could encourage meaningful change in Antarctic operations handling of plastic. For that I think the MS has merit.

I have significant misgivings about the title as I feel it is misleading.

The term "Atmospheric Dry Deposition" entails that the material was atmospherically entrained (the definition is >10 m AGL). I do not see any evidence of that. I think that the material merely underwent saltation transport from the beach. I.e. it's more likely it bounced along the ground up from the beach or from the Artigas base (a lot of red/white buildings). Evidencing the base as a source would have made a better focus. Aeolian transport is a more accurate description as it covers all manner of wind transport. I think all instances of "atmospheric dry deposition" should be removed before publication. This is not an atmospheric transport study and should not claim to be.

**Thank you very much for your comment. According your request, we have written: "*A pilot study about microplastics and mesoplastics in an Antarctic glacier: the role of aeolian transport*"**

Elevation and slope information is missing. The distance from the shoreline has not been made clear. One can infer from the maps, but distances and topography information should be clearly described. These are important to the understanding of possible transport vectors. A 3D elevation map can be made in GIS software.

**Following your request, we have written in section *2.1 Study area* "*The distance from the shoreline to Ionosferico lake is ~694 m.* […] *The distance from the shoreline to Uruguay lake is ~366 m.*". Regarding topography information, we have not found elevation maps of the area, except for the map already included in Figure 1B, 1C and 1D.**

Methodology: The 12hr testing is not useful as the material could have been blown in from just outside the squares or melting of the surface exposed new plastics (temperatures provided are from >300km away). It is disappointing that local meteo wind data from Artigas base was not included in the reporting. Average historical wind roses and wind reports from 322km away do not add to the evidence and I am sure the Artigas research station keeps daily records which would have been helpful. I also see no proof of the absence

 of rainfall/snowfall as I assume the authors did not camp on the glacier. I doubt that it would have any bearing on your results but again, being 322km away from a meteo monitoring station is not proof it did not rain/snow overnight on the glacier. I also note no mention of fragments being partially frozen into the ice which is hard to believe did not occur.

**Thank you very much for your comment. Meteorological data are not available from Artigas research station. However, we consider that local meteorological data from Villa Las Estrellas, which is located ~3.22 km (3220 meters) from Artigas base and keeps daily records, provides relevant evidence of wind direction and speed and absence of rain/snow during the study**

**period. We realized the distance is not clearly visible in Figure S1B, and we have now modified it.**

A 60% FT-IR match with the library with such massive pieces is at the extreme low end with the given settings. I would expect >80% (which is normally the lower limit) to be easily achievable. The Omnic software should have been able to tell you the composition of the other fragments too which would have been interesting and useful for locating sources. I suggest that there is something either wrong with the machines or the methods.

**Thank you very much for your comment. Recently, several studies used a 60% FT-IR match to identify airborne microplastics (doi.org/10.1016/j.scitotenv.2019.04.110, doi.org/10.1016/j.envint.2019.105127, doi.org/10.1016/j.jhazmat.2020.123223 or doi.org/10.1021/acs.est.9b03427). In our study, 75% of the pieces had percentages of FT-IR match with the library higher than 80%, and only two plastics a match < 70% (specifically 67% and 68 %). All the plastic spectra with matches between 67 - 79% FT-IR displayed significant peaks typical of plastics. Furthermore, the color and morphology of these plastics looked similar to the type of plastic they were ultimately classified into by the FT-IR analysis. Their low values in the matches could be consequence of several factors such as aging, dirt (see Figure S2) or humidity (e.g. some peak can variable from sample to sample as it depends on the amount of adsorbed water).**

I realise these are quite large fragments making contamination less likely, but "blanks" are an integral part of any microplastic study and the lack of even a lab blank is very disappointing. It should be mentioned what material the 100 mL ISO reagent bottles are made of (we can't assume the reader will know). I understand this MS started as a short piece with limited space but now it's an article it would also be useful to have the photos of the fragments in the supplementary. I note some grammatical errors throughout the MS and spelling mistakes which I assume will be addressed in final editorial (fig 2 D,F,H, *wavelength).

**Thank you very much for your comment. We have written wavenumber instead of wavelength in Figure 2 and have mentioned the material of the ISO reagent bottles. Furthermore, we have included representative photos of the plastics found (Figure S2).**

---

## Author Response (AR3)

Thank you very much for your comments. Following your suggestions, we have revised and improved the manuscript. We have accepted all changes and we have included the missing information.